# VitaBench: Benchmarking LLM Agents with Versatile Interactive Tasks in Real-world Applications

**Wei He**[1,2*†], **Yueqing Sun**[2*], **Hongyan Hao**[2*], **Xueyuan Hao**[2*], **Zhikang Xia**[2],
**Qi Gu**[2†], **Hui Su**[2], **Xunliang Cai**[2]
[1]Fudan University,   [2]Meituan LongCat Team

## ABSTRACT

As LLMs with agentic abilities are increasingly deployed in real-life scenarios, existing benchmarks fail to capture their inherent complexity of handling extensive information, leveraging diverse resources, and managing dynamic user interactions. To address this gap, we introduce **VitaBench**, a challenging benchmark that evaluates agents on versatile interactive tasks grounded in real-world settings. Drawing from daily applications in food delivery, in-store consumption, and online travel services, VitaBench presents agents with the most complex life-serving simulation environment to date, comprising 66 tools. Through a framework that eliminates domain-specific policies, we enable flexible composition of these scenarios and tools, yielding 100 cross-scenario tasks (main results) and 300 single-scenario tasks. Each task is derived from multiple real user requests and requires agents to reason across temporal and spatial dimensions, utilize complex tool sets, proactively clarify ambiguous instructions, and track shifting user intent throughout multi-turn conversations. Moreover, we propose a rubric-based sliding window evaluator, enabling robust assessment of diverse solution pathways in complex environments and stochastic interactions. Our comprehensive evaluation reveals that even the most advanced models achieve only 30% success rate on cross-scenario tasks, and less than 50% success rate on others. Overall, we believe VitaBench will serve as a valuable resource for advancing the development of AI agents in practical real-world applications.[1]

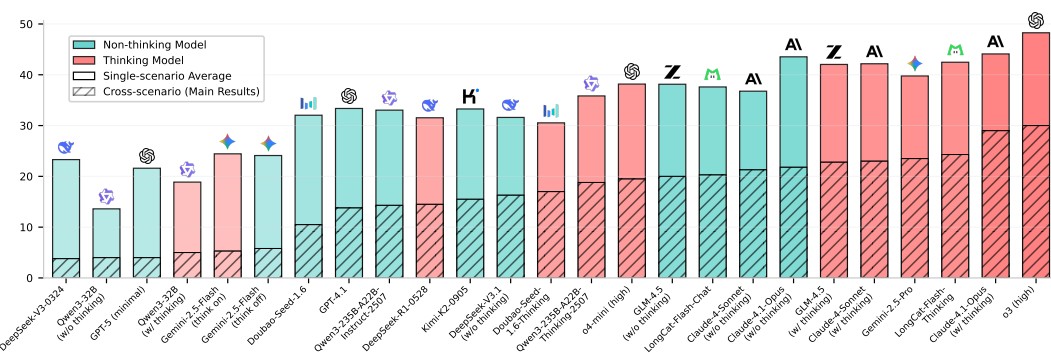

Figure 1: Overall performances on VitaBench, sorted by main results.

## 1 INTRODUCTION

Recent advances in large language models (LLMs) have significantly enhanced their complex reasoning and tool-use capabilities (Bai et al., 2025; Zeng et al., 2025; Li et al., 2025), leading to

---

[*]Equal contribution.

[†]Corresponding authors. Correspondence to: whe23@m.fudan.edu.cn, guqi03@meituan.com

[1]The code, dataset, and leaderboard are available at `https://vitabench.github.io`.

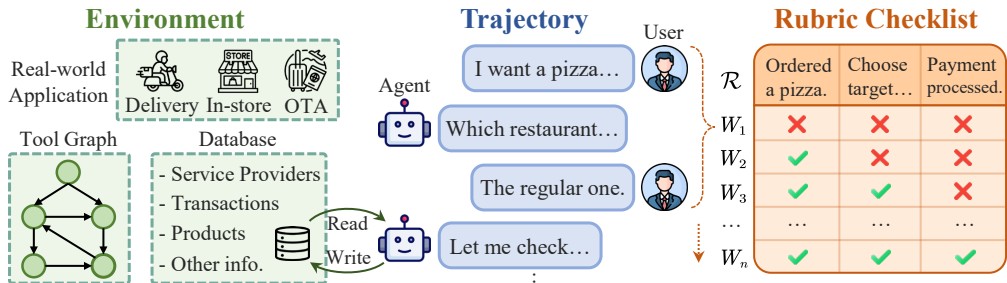

Figure 2: VitaBench sources tasks from real-world environments by composing interconnected tools, diverse user requests, and structured databases. Agents interact with users through multi-turn dialogue, while a rubric-based sliding-window evaluator tracks progress across the trajectory.

increased deployment of LLM agents in real-world applications. These improvements have simultaneously driven the evolution of agent-centric benchmarks (Yao et al., 2024; Barres et al., 2025; Lu et al., 2025), progressing from simple task execution to complex multi-turn interaction scenarios.

However, there remains a significant gap between controlled laboratory settings and real-world deployments that present inherently complex challenges. Early benchmarks (Qin et al., 2024; Patil et al., 2025) focused primarily on function-calling and parameter accuracy, introducing difficulty through increased tool counts or distractors, yet overlooking the intricate interdependencies between tools and their environments. Some recent efforts (Yao et al., 2024; Barres et al., 2025) have begun exploring real-world challenges, but often impose rigid domain-specific policies and constrained action spaces, overemphasizing instruction-following over autonomous exploration. Furthermore, many inadequately consider users as environmental components who bring inherent uncertainty, despite this being a critical challenge for practical agent applications (Qian et al., 2025).

This motivates our central research question:

*"What constitutes task complexity for agents in real-world applications?"*

Drawing inspiration from task complexity theories that examine structural, resource, and interaction dimensions (Liu & Li, 2012), we identify three fundamental aspects that shape agentic task complexity: (1) **reasoning complexity**, measured by the volume of environmental information that agents must process and integrate; (2) **tool complexity**, quantified through modeling tool sets as graphs based on inter-tool dependencies, where the node cardinality and edge density reflect the structural intricacy to navigate; (3) **interaction complexity**, characterized by the challenges arising from diverse user behavioral attributes and conversational patterns throughout multi-turn interaction.

Building on this framework, we present **VitaBench**[2] (short for Versatile Interactive Tasks Benchmark) to measure an agent's ability to handle the inherent complexity of real-world applications (overview in Figure 2). We construct 66 tools across three domains—delivery, in-store consumption, and online travel services—and model their intrinsic dependencies as a graph structure where policy information is inherently encoded. This allows agents to reason and explore autonomously without relying on domain-specific policies like $\tau$-bench (Yao et al., 2024). This design also enables flexible composition of scenarios and toolsets, facilitating the creation of 400 evaluation tasks spanning both single-scenario and cross-scenario settings. We derive each task from multiple authentic user requests and equip it with an independent environment containing annotated user profiles, spatiotemporal contexts, and comprehensive service databases. Given the extensive solution space of these instructions and environments where numerous valid pathways may exist, we introduce a rubric-based sliding window evaluator to assess the resulting long-horizon trajectories.

We evaluate multiple advanced LLMs on VitaBench, revealing that even the best-performing model achieves only 48.3% success rate across our 300 single-scenario tasks, with performance plummeting to 30.0% in cross-scenario settings where agents must navigate between different domain contexts and choose right tools from expanded action spaces (Figure 1). Our comprehensive analysis validates the three-dimensional complexity framework, showing strong correlations between complexity metrics and task difficulty across domains. Through systematic failure pattern analysis, we identify that reasoning errors dominate (61.8%), followed by tool usage errors (21.1%) and in-

---

[2]The name "Vita" derives from the Latin word for "Life", reflecting our focus on life-serving applications.

teraction management failures (7.9%), with agents exhibiting poor self-awareness and limited error recovery capabilities. Rigorous validation confirms the reliability of our evaluation components, establishing VitaBench as a challenging and reliable benchmark for advancing real-world agent capabilities. All code and data will be released to ensure reproducibility.

## 2 RELATED WORK

Table 1: Comparison of existing user interaction benchmarks across three complexity dimensions: reasoning, tool, and interaction. "✓" indicates fully addressed, "✓" indicates partially addressed, and "✗" indicates not addressed. Detailed explanations for each trait are provided in Appendix A.

| Benchmark | Reasoning Complexity | | | Tool Complexity | | | Interaction Complexity | | |
|---|---|---|---|---|---|---|---|---|---|
| | Multifaceted Information | Composite Objective | Goal Ambiguity | # Tools | Inter-tool Dependency | Cross Scenarios | # Turns (approx.) | User Profile | Behavior Attributes |
| ToolTalk (Farn & Shin, 2023) | ✗ | ✗ | ✗ | 28 | ✓ | ✗ | [2, 10] | ✗ | ✗ |
| IN3 (Qian et al., 2024) | ✗ | ✗ | ✓ | 0 | - | - | [2, 10] | ✗ | ✗ |
| MINT (Wang et al., 2024b) | ✗ | ✗ | ✓ | 8 | ✗ | ✗ | [2, 10] | ✗ | ✗ |
| ToolSandbox (Lu et al., 2025) | ✓ | ✗ | ✓ | 34 | ✓ | ✗ | [10, 30] | ✗ | ✗ |
| DialogTool (Wang et al., 2025) | ✓ | ✗ | ✗ | 31 | ✓ | ✓ | [10, 30] | ✓ | ✓ |
| UserBench (Qian et al., 2025) | ✓ | ✗ | ✓ | 5 | ✗ | ✗ | [10, 30] | ✓ | ✗ |
| $\tau$-Bench (Yao et al., 2024) | ✓ | ✗ | ✗ | 28 | ✓ | ✗ | [30, 50] | ✓ | ✗ |
| $\tau^2$-Bench (Barres et al., 2025) | ✓ | ✓ | ✗ | 38 | ✓ | ✗ | [30, 80] | ✓ | ✓ |
| VitaBench (ours) | ✓ | ✓ | ✓ | 66 | ✓ | ✓ | [50, 100] | ✓ | ✓ |

Early tool-use benchmarks (Huang et al., 2024; Qin et al., 2024; Patil et al., 2025) primarily focused on single-turn API calling accuracy, overlooking the inter-tool dependencies and dynamic interactions with users that characterize real-world applications. While recent work has recognized the need for evaluating advanced reasoning, tool manipulation, and interaction abilities, current benchmarks typically address these dimensions in isolation rather than comprehensively. Table 1 compares prominent agent-user interaction benchmarks across our proposed task complexity framework.

ToolTalk (Farn & Shin, 2023) first introduces multi-step tool execution through conversational interfaces but relies on predefined dialogue trajectories, limiting agent autonomy. While MINT (Wang et al., 2024b) emphasizes natural language feedback to guide agents and IN3 (Qian et al., 2024) focuses on detecting implicit intentions, both of them operate in relatively constrained agentic settings. More comprehensive frameworks like ToolSandbox (Lu et al., 2025) and the $\tau$-bench family (Yao et al., 2024; Barres et al., 2025) pioneer stateful execution and model tool interdependencies, yet constrain agents through verbose policies rather than allowing truly autonomous exploration. DialogTool (Wang et al., 2025) explores role-playing for engaging users but focuses primarily on agent-side capabilities, while UserBench (Qian et al., 2025) uniquely captures preference-driven interactions, though with limited task complexity otherwise. Several works (Yang et al., 2024; Wang et al., 2024a) also investigate agents' abilities to recognize incomplete conditions and proactively seek missing information. However, none of these benchmarks simultaneously challenge agents across multiple complexity dimensions. Our work aims to bridge this gap with VitaBench, which presents information-rich environments requiring agents to autonomously explore, dynamically interact with diverse users, and navigate intricate tool dependencies to address real-world demands.

## 3 VITABENCH: A BENCHMARK FOR VERSATILE INTERACTIVE TASKS

### 3.1 FORMULATION

**The POMDP Formalism.** We formalize the set of distinct environments as $\mathcal{E}$. For a specific environment $e \in \mathcal{E}$, we model the agent task as a partially observable Markov decision process (POMDP) $(\mathcal{U}, \mathcal{S}, \mathcal{A}, \mathcal{O}, \mathcal{T}, r)_e$ with instruction space $\mathcal{U}$, state space $\mathcal{S}$, action space $\mathcal{A}$, observation space $\mathcal{O}$, state transition function $\mathcal{T} : \mathcal{S} \times \mathcal{A} \to \mathcal{S}$, and reward function $r : \mathcal{S} \times \mathcal{A} \to \mathbb{R}$.

The agent interacts with both databases (through API tools) and a simulated user. Accordingly, the action space $\mathcal{A}$ consists of two types of actions: tool invocation and interactive dialogue with the user. The state space $\mathcal{S}$ comprises the state of the database and the user state, i.e., $\mathcal{S} = \mathcal{S}_{\text{db}} \otimes \mathcal{S}_{\text{user}}$. The observation space $\mathcal{O}$ includes the database feedback after tool calls and the conversation history

with the user, i.e., $\mathcal{O} = \mathcal{O}_{db} \otimes \mathcal{O}_{user}$. The state transition function $\mathcal{T}$ decomposes accordingly: API calls follow deterministic transitions $\mathcal{T}_{db}$ implemented as Python functions, while user interactions follow stochastic transitions $\mathcal{T}_{user}$ implemented using a language model.

Given an instruction $u \in \mathcal{U}$, the initial state $s_0$ represents the token sequences of the given prompt and the initial state of the database. The agent receives an initial observation $o_0 \in \mathcal{O}$, which typically includes the first-round user request and the available tool sets. The LLM agent, parameterized by $\theta$, generates an action $a_1 \sim \pi_\theta(\cdot|o_0)$ based on its policy $\pi_\theta$. Subsequently, the state transitions to $s_1 \in \mathcal{S}$, and the agent receives feedback $o_1 \in \mathcal{O}$. At each step $t$, the agent acts based on the current observable history, which can be denoted as $(o_0, a_1, o_1, \ldots, a_{t-1}, o_{t-1})$, generating action $a_t \sim \pi_\theta(\cdot|o_0, a_1, o_1, \ldots, a_{t-1}, o_{t-1})$. The agent continues interacting with the environment until the task is completed or the maximum number of steps is reached. From the environment's perspective, the complete state transition trajectory can be represented as:

$$\tau = (s_0, a_1, s_1, a_2, s_2, \ldots, a_T, s_T) \sim \pi_\theta(\tau|e, u), \tag{1}$$

where $T$ denotes the total number of interaction rounds. Note that the trajectory $\tau$ captures the complete state transitions, while the agent only has access to partial observations $o_t$ derived from states $s_t$. The reward $r(e, u, \tau) \in [0, 1]$ is computed after the interaction ends.

**Agentic Task Complexity Framework.** Building upon the POMDP formalism and drawing inspiration from multi-perspective complexity frameworks (Liu & Li, 2012), we formalize task complexity along three dimensions that capture the challenges agents face in real-world applications:

$$\mathcal{C}_{task} = \langle \mathcal{C}_{reason}, \mathcal{C}_{tool}, \mathcal{C}_{interact} \rangle. \tag{2}$$

- **Reasoning complexity** $\mathcal{C}_{reason}$ quantifies the cognitive demands of processing extensive environmental information under partial observability. We characterize this through the entropy of the observation space $H(\mathcal{O})$ and the degree of partial observability $\eta = 1 - \frac{|\mathcal{O}|}{|\mathcal{S}|}$, where higher values indicate greater uncertainty in state estimation. Building on this framework, we construct large-scale databases and composite tasks with multiple explicit and implicit reasoning points.

- **Tool complexity** $\mathcal{C}_{tool}$ captures the structural intricacy of navigating interconnected action spaces. We model the toolset as a directed graph $G = (V, E)$ where vertices represent individual tools and edges encode inter-tool dependencies. Complexity emerges from graph cardinality $|V|$, edge density $\rho = \frac{|E|}{|V|(|V|-1)}$, the coverage ratio $\frac{|V_{task}|}{|V|}$ of task-relevant subgraph. Cross-scenario settings further amplify this by expanding the action space $\mathcal{A}$ across multiple domains.

- **Interaction complexity** $\mathcal{C}_{interact}$ reflects the challenges of managing dynamic multi-turn conversations with users. User profiles encode personal attributes (e.g., gender, age, dietary restrictions) that influence task requirements. Behavior attributes introduce variability in cooperation levels and goal ambiguity, necessitating proactive clarification. Moreover, real-world users exhibit dynamic states $\mathcal{S}_{user}$ that evolve throughout the interaction, requiring continuous strategy adaptation.

## 3.2 BENCHMARK CONSTRUCTION

We construct VitaBench through a systematic pipeline illustrated in Figure 3. Specifically, this process can be divided into two stages:

**Stage I: Framework Design.** We construct VitaBench through systematic abstraction of real-world life-serving scenarios across three domains: *Delivery* (food and product delivery), *In-store Consumption* (dining and other services), and *Online Travel Agency (OTA)* (hotel bookings, attraction reservations, flight and train ticket management). By referencing existing application implementations, we derive simplified API tools that capture essential functionalities. We model inter-tool dependencies as a directed graph $G = (V, E)$ and augment tool descriptions with preconditions (states required before execution) and post-conditions (expected outcomes after execution). This graph-based design naturally encodes domain rules into tool structures, eliminating the need for verbose policy documents while simultaneously increasing reasoning complexity and facilitating cross-domain composition. For instance, `modify_order` requires prior execution of `get_order_detail` to obtain necessary information, reflecting natural workflow dependencies. The complete toolset used in VitaBench is documented in Appendix B.

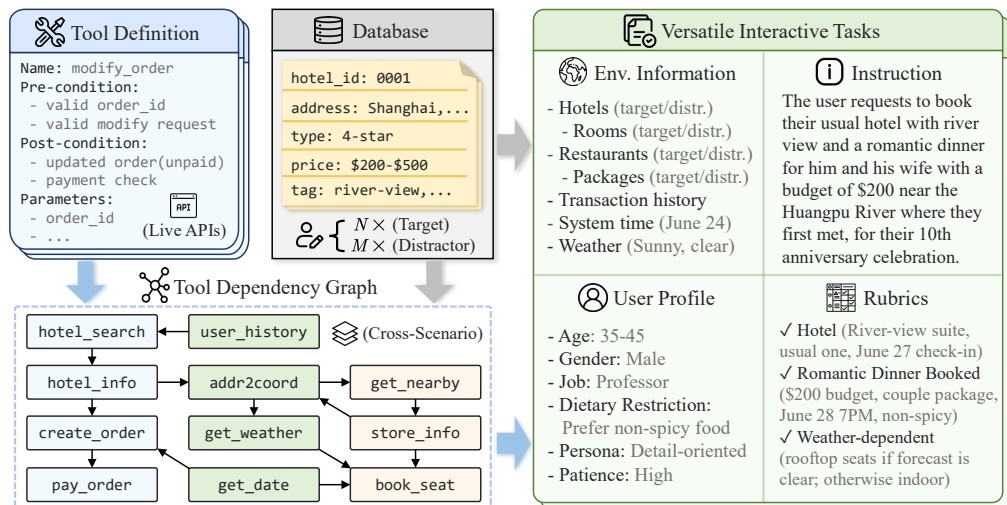

Figure 3: Overview of the VitaBench construction pipeline and a simplified cross-scenario example.

To capture the inherent uncertainty in real-world interactions, we implement a user simulator following Yao et al. (2024). The simulator receives complete instructions containing multiple requirements but reveals them progressively to agents, and provides implicit constraints only upon inquiry. We configure each simulated user with unique profiles and behavioral attributes, employing prompt-based constraints to maintain persona consistency while minimizing critical errors that would impede task completion (validated in Section 5.1). Since fully unconstrained user behavior would introduce excessive randomness, our simulator adopts controlled design to provide a balance between realism and evaluative stability, enabling fair and reproducible comparisons across models. Note that while user profiles are accessible to agents, we establish knowledge boundaries to reflect realistic scenarios—for example, agents cannot directly access dietary restrictions but must infer them from order history or user responses.

**Stage II: Task Creation.** Our data collection pipeline consists of four components: user profiles, task instructions, environmental information, and rubrics. User profiles derive from authentic platform data, which we anonymize and enrich to create distinct personas with varied personal attributes and communication styles. These attributes encompass emotional expressions (e.g., impatient, anxious, indifferent) and interaction patterns (e.g., detail-oriented, dependent, logical), leading to diverse conversational dynamics throughout multi-turn dialogues. Task instructions synthesize multiple real user requests into composite objectives, which we manually review and refine to ensure clarity and feasibility. Instructions either coordinate multiple sub-goals within a single domain or span across different domains in cross-scenario settings, requiring agents to navigate between distinct contexts. For environmental data, we combine service provider and product information from real-world life-serving platforms with model-generated synthetic augmentation under human supervision. We deliberately intermix target options that satisfy all constraints with distractor options that violate specific requirements, creating extensive search spaces with numerous candidates while maintaining only a handful of valid solutions per task. Additionally, we generate transaction histories to support requirements involving consumption patterns (e.g., "*order the same meal as last time*" or "*book my usual hotel*"). We iteratively refine each task through multiple trials with human verification, eliminating ambiguities while preserving multiple valid solution pathways. Through this process, we construct 400 tasks with comprehensive databases detailed in Table 2, where individual tasks typically involve 5-20 service providers and can include over 100 products in certain cases.

Table 2: Data statistics of VitaBench.

| | Cross-Scen. | Delivery | In-store | OTA |
|---|---|---|---|---|
| **Databases** | | | | |
| Service Providers | 1,324 | 410 | 611 | 1,437 |
| Products | 6,946 | 788 | 3,277 | 9,693 |
| Transactions | 447 | 48 | 28 | 154 |
| **API Tools** | | | | |
| Write | 27 | 4 | 9 | 14 |
| Read | 33 | 10 | 10 | 19 |
| General | 6 | 6 | 5 | 5 |
| **Tasks** | 100 | 100 | 100 | 100 |

### 3.3 RUBRIC-BASED SLIDING WINDOW EVALUATOR

Evaluating long-form agent trajectories presents unique challenges due to their extensive length and multiple valid solution paths. While Yao et al. (2024) rely on predefined database state comparisons, such methods cannot capture nuanced requirements such as recommendations or planning behaviors that leave final states unchanged, nor provide supervision for intermediate transitions. Recent rubric-based evaluation methods (Arora et al., 2025; Ruan et al., 2025) inspire our approach by decomposing complex goals into atomic criteria, enabling comprehensive requirement coverage. With carefully-designed rubrics, LLM-as-a-Judge can effectively replace fine-grained human judgments while maintaining high accuracy. To address the challenge that multi-turn trajectories often exceed context lengths, we propose a sliding window evaluator that processes trajectories in sequential segments while maintaining continuity through persistent rubric state tracking.

We manually design rubrics $\mathcal{R} = \{r_1, \ldots, r_k\}$ for each task, comprising atomic criteria derived from task information (e.g., "*restaurant within 500m*", "*user only eats vegetarian food*"). Each trajectory is divided into overlapping windows $W_i$ of $w$ consecutive turns, with adjacent windows sharing $\delta$ turns to ensure information coherence. When processing each window, the evaluator extracts rubric-relevant information and propagates it forward to enable consistent cross-window judgments. The evaluator maintains a state vector $\mathbf{s} \in {0, 1}^k$ that persistently records criterion satisfaction across windows—once a rubric item $r_j$ is satisfied in any window, $s_j$ is marked as $1$, and if a previously satisfied criterion is later negated, the corresponding state will be reset to $0$ accordingly. For benchmark evaluation, we adopt a strict all-or-nothing scoring where success requires satisfying all rubric items: score $= \mathbb{1}[\sum_j s_j = k]$. Nevertheless, the fine-grained rubrics enable detailed scoring analysis for identifying trajectory differences, providing valuable dense signals for reinforcement learning. Human evaluation yields strong inter-rater agreement with Cohen's $\kappa \geq 0.81$ (Cohen, 1960) as shown in Section 5.1, validating the reliability of our approach.

## 4 EXPERIMENTS

### 4.1 EXPERIMENTAL SETUPS

**Models.** We evaluate various state-of-the-art proprietary and open language models for agents. The complete list of evaluated models is provided in Appendix C. The leaderboard is divided into thinking and non-thinking model categories. For hybrid models that support toggling between two modes, we evaluate the think-on and think-off configuration in two categories.

**Methods.** The language agents are implemented as function-calling agents, with all tools provided in the OpenAI tool schemas. Note that we use the minimal agent design because our focus is to evaluate the intrinsic capabilities of foundation models without introducing confounding factors from rapidly evolving agentic systems. We do not limit the number of interaction rounds for agent models, and the task terminates when the agent outputs "###STOP###" or encounters a failure. The user simulator is implemented using `gpt-4.1-2025-04-14`. The evaluator is implemented using `claude-3.7-sonnet` to avoid overlap with the evaluated agent models. For main results, each task is run four times with a consistent LLM temperature of $0.0$ to promote deterministic outputs. The prompt templates we used for agent, user and evaluator are detailed in Appendix E. An analyse of representative cost and latency statistics is provided in Appendix D.

**Metrics.** For the results from four runs, we report Avg@$4$, Pass@$4$, and Pass^$4$ metrics averaged across tasks. Pass@$k$ represents the probability that at least one out of $k$ i.i.d. task trials is successful. Pass^$k$ represents the probability that all $k$ i.i.d. task trials are successful (Yao et al., 2024).

### 4.2 MAIN RESULTS

Table 3 presents comprehensive evaluation results on VitaBench. We can observe that:

**Real-world tasks pose great challenges for current agents.** Performance varies significantly across domains and correlates strongly with environmental complexity. Cross-scenario tasks expose

Table 3: Performance comparison of non-thinking and thinking models across different domains.

| Models | Cross-Scenarios | | | Delivery | | | In-store | | | OTA | | |
|---|---|---|---|---|---|---|---|---|---|---|---|---|
| | Avg @4 | Pass @4 | Pass ^4 | Avg @4 | Pass @4 | Pass ^4 | Avg @4 | Pass @4 | Pass ^4 | Avg @4 | Pass @4 | Pass ^4 |
| *Non-thinking Models* | | | | | | | | | | | | |
| DeepSeek-V3-0324 | 3.8 | 12.0 | 0.0 | 25.3 | 53.0 | 5.0 | 34.3 | 71.0 | 5.0 | 10.3 | 26.0 | 1.0 |
| Qwen3-32B (w/o thinking) | 4.0 | 12.0 | 0.0 | 16.5 | 37.0 | 3.0 | 21.3 | 47.0 | 2.0 | 3.0 | 11.0 | 0.0 |
| GPT-5 (minimal) | 4.0 | 9.0 | 0.0 | 30.0 | 64.0 | 6.0 | 27.0 | 60.0 | 2.0 | 7.8 | 22.0 | 0.0 |
| Gemini-2.5-Flash (think off) | 5.8 | 17.0 | 1.0 | 31.0 | 65.0 | 6.0 | 22.8 | 46.0 | 3.0 | 18.5 | 44.0 | 1.0 |
| Doubao-Seed-1.6 | 10.5 | 29.0 | 0.0 | 37.8 | 65.0 | 12.0 | 39.5 | 73.0 | 9.0 | 18.8 | 39.0 | 3.0 |
| GPT-4.1 | 13.8 | 35.0 | 0.0 | 37.8 | 67.0 | 11.0 | 42.5 | 71.0 | 17.0 | 19.8 | 42.0 | 1.0 |
| Qwen3-235B-A22B-Instruct-2507 | 14.3 | 38.0 | 0.0 | 34.3 | 66.0 | 6.0 | 44.8 | 87.0 | 13.0 | 20.0 | 45.0 | 1.0 |
| Kimi-K2-0905 | 15.5 | 39.0 | 2.0 | 35.3 | 68.0 | 9.0 | 42.5 | 78.0 | 10.0 | 22.0 | 46.0 | 4.0 |
| DeepSeek-V3.1 (w/o thinking) | 16.3 | 40.0 | 1.0 | 34.0 | 67.0 | 6.0 | 42.5 | 76.0 | 7.0 | 18.3 | 47.0 | 1.0 |
| GLM-4.5 (w/o thinking) | 20.0 | 47.0 | 1.0 | 45.8 | 72.0 | **20.0** | 48.3 | 82.0 | 13.0 | 20.3 | 45.0 | 2.0 |
| LongCat-Flash-Chat | 20.3 | 45.0 | 2.0 | 39.5 | 71.0 | 15.0 | 50.5 | 84.0 | 15.0 | 22.8 | 49.0 | 2.0 |
| Claude-4-Sonnet (w/o thinking) | 21.3 | **49.0** | **4.0** | 39.0 | 69.0 | 17.0 | 46.3 | 78.0 | 10.0 | 25.0 | 49.0 | 7.0 |
| Claude-4.1-Opus (w/o thinking) | **21.8** | 47.0 | 3.0 | **46.0** | **78.0** | 13.0 | **53.8** | **85.0** | 21.0 | 30.8 | 60.0 | **9.0** |
| *Thinking Models* | | | | | | | | | | | | |
| Qwen3-32B (w/ thinking) | 5.0 | 24.0 | 0.0 | 22.8 | 53.0 | 4.0 | 26.5 | 60.0 | 3.0 | 7.3 | 18.0 | 1.0 |
| Gemini-2.5-Flash (think on) | 5.3 | 14.0 | 0.0 | 32.0 | 62.0 | 9.0 | 23.0 | 57.0 | 3.0 | 18.3 | 39.0 | 1.0 |
| DeepSeek-R1-0528 | 14.5 | 39.0 | 0.0 | 40.3 | 72.0 | 11.0 | 41.3 | 79.0 | 7.0 | 13.0 | 32.0 | 2.0 |
| Doubao-Seed-1.6-Thinking | 17.0 | 42.0 | 1.0 | 30.3 | 59.0 | 10.0 | 43.3 | 78.0 | 10.0 | 18.0 | 45.0 | 2.0 |
| Qwen3-235B-A22B-Thinking-2507 | 18.8 | 45.0 | 2.0 | 44.0 | 78.0 | 9.0 | 46.0 | 80.0 | 9.0 | 17.5 | 41.0 | 2.0 |
| o4-mini (high) | 19.5 | 49.0 | 1.0 | 44.5 | 80.0 | 15.0 | 46.5 | 81.0 | 15.0 | 23.5 | 50.0 | 5.0 |
| GLM-4.5 (w/ thinking) | 22.8 | 48.0 | 2.0 | 44.5 | 77.0 | 14.0 | 52.8 | 80.0 | 22.0 | 28.8 | 55.0 | 7.0 |
| Claude-4-Sonnet (w/ thinking) | 23.0 | 51.0 | **6.0** | 46.0 | 78.0 | 15.0 | 51.5 | 80.0 | 21.0 | 29.0 | 55.0 | 9.0 |
| Gemini-2.5-Pro | 23.5 | 53.0 | 5.0 | 49.0 | 81.0 | 16.0 | 43.8 | 78.0 | 12.0 | 26.5 | 54.0 | 6.0 |
| LongCat-Flash-Thinking | 24.3 | 54.0 | 3.0 | 42.3 | 71.0 | 13.0 | **56.8** | 85.0 | **25.0** | 28.3 | 59.0 | 6.0 |
| Claude-4.1-Opus (w/ thinking) | 29.0 | 56.0 | **6.0** | 47.5 | 80.0 | 17.0 | 52.5 | 78.0 | 20.0 | 32.3 | 57.0 | 9.0 |
| o3 (high) | **30.0** | **61.0** | **6.0** | **53.5** | **83.0** | **24.0** | 53.5 | **86.0** | 19.0 | **37.8** | **66.0** | **10.0** |

the most severe limitations: even top-performing models achieve only $30.0\%$ Avg@4 score, compared to over $50\%$ in single-domain settings. This dramatic gap reveals fundamental deficiencies in navigating expanded action spaces and coordinating across distinct domains. Notably, task difficulty does not correlate with database scale—the in-store domain, despite having far more products, proves easier than delivery settings. This counterintuitive finding shows how real-world complexity emerges: delivery tasks demand precise coordination of multiple items under strict constraints, while in-store operations remain straightforward despite larger candidate pools.

**Exploration improves performance but reveals stability issues.** The Pass@$k$ and Pass^$k$ metrics capture complementary aspects of model behavior. Pass@4 results show that increased sampling substantially improves completion rates, indicating that complex environments reward exploration, which suggests promising directions for RL approaches. However, Pass^4 metrics reveal concerning instability, with even top models dropping to near-zero consistency rates. To further validate this observation, we evaluate representative models with $k = 32$ samples (Figure 4), confirming that while exploration yields marginal gains, fundamental stability challenges persist even for leading agentic models like Claude-4-Sonnet.

**Thinking mechanisms improve both effectiveness and efficiency.** Thinking models generally outperform their non-thinking versions, with improvements such as Claude-4.1-Opus increasing from $21.8\%$ to $29.0\%$ and GLM-4.5 from $20.0\%$ to $22.8\%$. Moreover, thinking mechanisms lead to efficiency improvements, as shown in Figure 5 where thinking models tend to achieve better performance with fewer turns on average. For instance, the overall trend demonstrates that higher-performing models require fewer interaction turns, with thinking models achieving an average performance of $23.8\%$ compared to $17.9\%$ for non-thinking models, while maintaining comparable turn counts (61.1 vs 69.9 turns respectively). This efficiency gain stems from two factors: the better decomposition of complex multi-step plans and more targeted user interactions through precise clarifying questions.

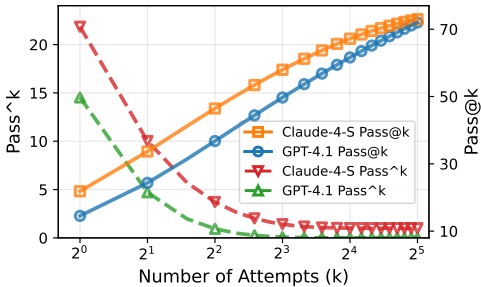

Figure 4: Pass@$k$ vs. Pass^$k$ performance.

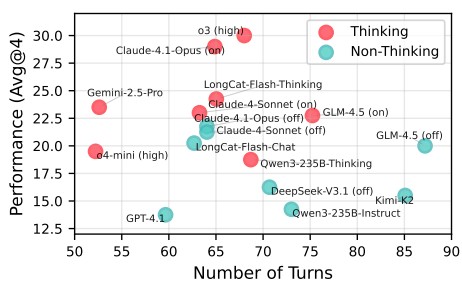

Figure 5: Model performance vs. Turns.

## 5 DISCUSSION

### 5.1 RELIABILITY ANALYSIS OF VITABENCH COMPONENTS

Given that our benchmark incorporates model-based components for user simulation and trajectory evaluation, we conduct reliability analyses to validate their effectiveness and stability.

**Reliability of user simulator.** We evaluate our user simulator across two critical dimensions: information fidelity and persona consistency. For information fidelity, two annotators assess 100 conversations examining adherence to task instructions and user profiles, absence of hallucinations, and contextual relevance. As shown in Figure 6(a), the simulator achieves high fidelity with $9.48/10$ average score across all scenarios. Minor deviations manifest as nat-

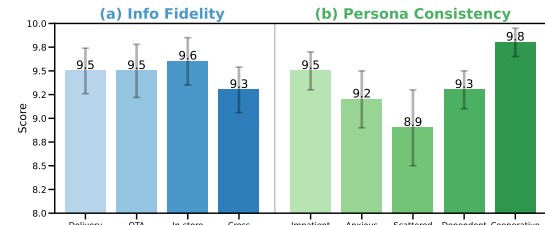

Figure 6: User simulator reliability evaluation.

ural conversational variations (e.g., "*cannot eat spicy*" vs. "*prefer non-spicy food*") that enhance dialogue authenticity without compromising task requirements. Notably, the simulator appropriately responds "*I don't know*" when queried about unprovided information, maintaining strict source fidelity. For persona consistency, we test five distinct personality types across 100 conversations, measuring behavioral alignment through language style, decision patterns, and emotional expressions. Figure 6(b) demonstrates strong persona-behavior alignment averaging $9.34/10$. Cooperative personas exhibit the highest consistency, aligning with LLMs' inherent collaborative tendencies, while scattered personas show lower controllability.

To further examine whether the user simulator implicitly favors agents with similar behaviors or reasoning patterns, we conduct a cross-model comparison by replacing GPT-4.1 with Claude-4-Sonnet as the simulator in cross-scenario tasks. As shown in Table 4, GPT-4.1's performance remains nearly unchanged under different simulators, whereas Claude-4-Sonnet exhibits a mild drop when interacting with a simulator of the same family. It indicates that any implicit cooperation between the simulator and the agent is minimal.

Table 4: Cross-model analysis of potential simulator–agent cooperation.

| User Sim. | Agent | Evaluator | Avg@4 | Pass@4 | Pass^4 |
|---|---|---|---|---|---|
| GPT-4.1 | GPT-4.1 | Claude-3.7-S | 13.8 | 35.0 | 0.0 |
| GPT-4.1 | Claude-4-S | Claude-3.7-S | 21.3 | 49.0 | 4.0 |
| Claude-4-S | GPT-4.1 | Claude-3.7-S | 13.8 | 34.0 | 0.0 |
| Claude-4-S | Claude-4-S | Claude-3.7-S | 19.5 | 45.0 | 5.0 |

**Reliability of evaluator.** We conduct ablation experiments to validate our rubric-based sliding window evaluator on GLM-4.5's cross-scenario trajectories. Table 5 compares four configurations against human-annotated ground truth: (1) baseline with sliding window and rubric, (2) full trajectory with rubric, (3) sliding window without rubric, and (4) full trajectory without rubric. For configuration (3), we employ external memory module to main-

Table 5: Ablation study of evaluator components. The "Score" refers to the evaluation score assigned to corresponding GLM-4.5 trajectories after applying the respective method.

| Method | Score | Task Acc. | Rubric Acc. | Cohen's $\kappa$ |
|---|---|---|---|---|
| Baseline | 20.0 | **95.0** | **88.5** | **0.828** |
| w/o Sliding Window | 19.0 | 90.0 | 87.6 | 0.604 |
| w/o Rubric Checklist | 91.0 | 22.0 | - | 0.018 |
| w/o Both | 82.0 | 32.0 | - | 0.067 |

tain context awareness. The result shows that our proposed method achieves the highest agreement with human judgments (Cohen's $\kappa = 0.828$), significantly outperforming methods without rubric structure ($\kappa < 0.07$). While full trajectory with rubric yields similar final scores (19% vs. 20%), the evaluation model's limited long-context capability hinders accurate assessment of all rubrics in the full trajectory. The sliding window design effectively handles this while maintaining 95% task-level accuracy, confirming the reliability of our approach.

Since the evaluator is also an LLM, a natural concern is whether its reasoning patterns may align with certain agent families. To mitigate this, all rubric items are designed to be objective, fine-grained, and binary (0/1), minimizing dependence on phrasing or evaluator-specific preferences.

**Statistical reliability of evaluation.** Beyond the aforementioned components, evaluation reliability is further affected by inherent agent stochasticity. Despite setting temperature to 0.0, cumulative perturbations in multi-turn interactions amplify into divergent trajectories. To determine the optimal number of evaluation runs, we conduct resampling analysis based on 32 independent trials. For each $k \in [1, 20]$, we calculate the Mean Squared Error (MSE) of $k$-run average estimates relative to the expected value (32-run average) by sampling different $k$-combinations from the 32 trials. Figure 7 demonstrates that $k = 4$ runs achieve optimal balance between statistical precision and computational cost. Compared to $k = 1$, using $k = 4$ reduces MSE by 77.5%, while increasing to $k = 8$ only provides marginal reduction despite doubling computational overhead. So we choose 4 evaluation runs for the main experiments.

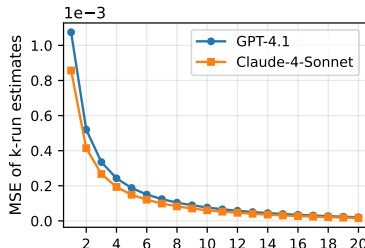

Figure 7: MSE stability across different evaluation run counts.

## 5.2 TASK COMPLEXITY ANALYSIS

**Reasoning and Tool Complexity.** We analyze how reasoning complexity $\mathcal{C}_{\text{reason}}$ and tool complexity $\mathcal{C}_{\text{tool}}$ affect task difficulty. Table 6 summarizes complexity characteristics and performance across four domains. Reasoning complexity depends on both the number of reasoning points and search space size. Cross-scenario and OTA tasks require 10.3 and 9.7 reasoning points respectively, demanding complex inference under partial observability. Despite having the largest search space, the In-store domain achieves the high-

Table 6: Environmental complexity characteristics and performance analysis.

| Domain | Performance | Reasoning Complexity | | Tool Complexity | | |
|---|---|---|---|---|---|---|
| | All Models | Reas. Pts. | Search Space | Tools | Edges | Density |
| In-store | 42.1 | 5.6 | 3,916 | 24 | 68 | 12.3% |
| Delivery | 38.0 | 7.4 | 1,246 | 20 | 50 | 13.2% |
| OTA | 20.7 | 9.7 | 11,284 | 38 | 309 | 22.0% |
| Cross-scenario | 16.2 | 10.3 | 8,717 | 66 | 512 | 11.2% |

est performance (42.1%) due to fewer reasoning points. Tool complexity strongly correlates with task difficulty: Cross-scenario tasks, with the highest tool complexity (66 tools, 512 dependency edges), yield the lowest performance (16.2%). The OTA domain's 22% graph density indicates complex inter-tool dependencies, resulting in poor performance (20.7%).

**Interactive Complexity.** We conduct ablation studies to quantify interaction complexity $\mathcal{C}_{\text{interact}}$, evaluating two models under three conditions: (1) our default user simulator with full persona and behavioral attributes, (2) user simulator without these attributes (neutral user), and (3) solo agent setting where complete instructions are provided upfront without user interaction.

As shown in Figure 8, user interaction introduces substantial complexity beyond direct task execution. The performance gap between default and neutral users is relatively small for Claude-4-Sonnet compared to GPT-4.1-Mini, suggesting that conversational styles primarily challenge weaker models. Conversely, Claude-4-Sonnet gains more in solo agent mode, indicating that it excels at processing complex instructions in a single round. These findings validate interaction complexity as a fundamental dimension of task difficulty, with its impact varying significantly based on model capabilities.

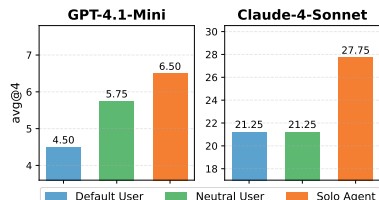

Figure 8: Ablation study of user simulation configurations.

To further assess whether specific simulated user attributes alter task difficulty, we additionally evaluate Claude-4-Sonnet on 100 cross-scenario tasks while fixing the user simulator to one of five predefined personas. As shown in Table 7, performance exhibits moderate variation: cooperative users lead to slightly higher success rates, whereas anxious or scattered personas reduce both performance and stability. These results complement the ablation in Figure 8, confirming that interaction complexity arises not only from the presence of user communication, but also from differences in user styles.

Table 7: Performance of Claude-4-Sonnet under fixed user personas.

| Persona | Avg@4 | Pass@4 | Pass^4 |
|---|---|---|---|
| Random | 21.3 | 49.0 | 4.0 |
| Impatient | 21.5 | 48.0 | 4.0 |
| Anxious | 18.5 | 41.0 | 2.0 |
| Scattered | 19.3 | 47.0 | 0.0 |
| Dependent | 20.6 | 45.0 | 3.0 |
| Cooperative | 22.8 | 50.0 | 5.0 |

### 5.3 ERROR PATTERN ANALYSIS IN VITABENCH

To understand the failure modes of current agents on VitaBench, we analyze cross-scenario task trajectories from Claude-4.1-Opus, categorizing 76 failed rubrics into distinct error patterns.

We classify the failures into three main categories aligned with our agentic task complexity framework, as illustrated in Figure 9. **Reasoning errors** ($61.8\%$) dominate the failure landscape, revealing fundamental limitations in task decision-making and handling composite objectives with multiple constraints. **Tool-use errors** ($21.1\%$) stem from incorrect tool selection, parameter passing mistakes, and inability to recover from invocation failures. **Interaction errors** ($7.9\%$) reflect challenges in dialogue management, where agents fail to proactively clarify ambiguous requirements and lose

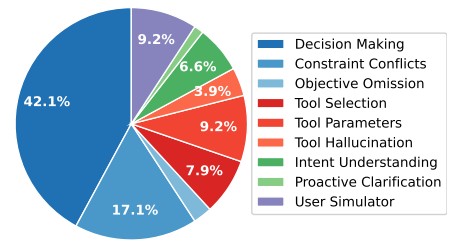

Figure 9: Error distribution of VitaBench.

track of user preferences across extended conversations. The remaining $9.2\%$ are user simulator errors, an inherent stochastic behavior that we mitigate through multiple runs (Yao et al., 2024).

From these failures, we identify several recurring patterns that highlight weaknesses in current agents. First, complex reasoning failures occur systematically across spatial-temporal and common-sense reasoning, indicating limited ability to integrate knowledge across multi-faceted information. Second, agents exhibit poor self-awareness of their capabilities, frequently abandoning tasks despite having access to appropriate tools, revealing fundamental gaps in understanding their own action boundaries. Third, agents show limited error recovery when facing tool failures or unclear user responses, with most repeating failed attempts rather than adapting other strategies.

## 6 CONCLUSION

In this work, we rethink the evaluation of LLM-based agents through the lens of real-world task complexity, introducing **VitaBench** to bridge the gap between controlled benchmarks and practical deployments. By formalizing agentic task complexity across reasoning, tool use, and interaction dimensions, VitaBench provides the most intricate life-serving simulation environment to date with 66 tools and 400 tasks spanning single- and cross-scenario settings. Our evaluation reveals that even advanced models achieve only $30.0\%$ success rate under cross-scenario settings (main result) and less than $50\%$ success rate under single-scenario settings. We believe VitaBench offers a challenging testbed and actionable insights for advancing real-world agent applications.

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

## ETHICS STATEMENT

Our research fully complies with the ICLR Code of Ethics. We obtained all datasets in compliance with relevant usage guidelines, ensuring no violation of privacy. Throughout our investigation, we carefully avoided any bias or discriminatory effects. Our work excluded any personally identifiable data and avoided procedures that might compromise privacy or security. We remain dedicated to upholding transparency and research integrity at every stage.

## REPRODUCIBILITY STATEMENT

We have prioritized making our findings fully reproducible. We have included our complete codebase and datasets in the supplementary materials and will make them fully available to the public, enabling independent validation and replication. The experimental setup is described in detail in the paper. We believe these provisions should empower the research community to reproduce our work and further advance the field.

## LLM USAGES STATEMENT

LLMs are utilized in this manuscript for partial grammatical checks and language polishing. The authors are fully responsible for the final content.

## A   COMPARISON TRAITS DETAILS

We identify nine traits across three complexity dimensions that characterize related benchmarks.

- **Multifaceted Information**: Tasks require integrating temporal and spatial information, common-sense knowledge, and understanding of various environmental components to form coherent solutions.
- **Composite Objective**: Tasks involve multiple interdependent sub-goals derived from user requirements that must be coordinated across different aspects (e.g., booking flights, hotels, and activities within budget constraints).
- **Goal Ambiguity**: User inputs may be underspecified or vague, requiring agents to proactively seek clarification, infer missing information, or iteratively refine their understanding through dialogue.
- **# Tools**: The number of distinct tools or APIs available. Larger tool inventories increase selection complexity and require understanding diverse functionalities.
- **Inter-tool Dependency**: Tools exhibit dependencies through pre-conditions (states required before execution) and post-conditions (outcomes after execution), requiring agents to plan multi-step execution strategies.
- **Cross Scenarios**: The benchmark design enables flexible composition of tools across multiple domains, requiring agents to navigate between distinct contexts rather than relying on domain-specific patterns.
- **# Turns**: The approximate number of trajectory turns required. Longer trajectories test context maintenance and handling of progressively revealed information throughout multi-turn conversations.
- **User Profile**: Persistent user profiles encode personal attributes (e.g., age, gender) and preferences that influence task requirements (e.g., dietary restrictions), necessitating personalized agent responses.
- **Behavior Attributes**: Modeling diverse user behavioral patterns including emotional expressions (e.g., impatient, anxious), interaction patterns (e.g., detail-oriented, dependent), and dynamic engagement levels based on agent performance such as reduced willingness to respond when receiving repetitive answers.

## B   COMPLETE TOOL LIST IN VITABENCH

VitaBench provides a comprehensive toolset that abstracts the APIs required in three real-world scenarios. Each tool is implemented as a Python function, and all returned values are drawn from a database in the equipped sandbox, ensuring consistency and determinism across executions. The complete toolset consists of 66 tools, listed below.

---

**Complete Tool List in VitaBench**

**Delivery (12):**
```
delivery_distance_to_time, get_delivery_store_info,
get_delivery_product_info, delivery_store_search_recommand,
delivery_product_search_recommand, create_delivery_order,
pay_delivery_order, get_delivery_order_status,
cancel_delivery_order, modify_delivery_order,
search_delivery_orders, get_delivery_order_detail
```

**In-store (16):**
```
instore_shop_search_recommend, instore_product_search_recommend,
create_instore_product_order, search_instore_reservation,
instore_cancel_order, instore_book, pay_instore_book,
instore_cancel_book, instore_reservation,
```

---

```
instore_modify_reservation, instore_cancel_reservation,
get_instore_orders, get_instore_reservations,
get_instore_books, search_instore_book, pay_instore_order
```

**OTA (30):**
```
get_ota_hotel_info, get_ota_attraction_info,
get_ota_flight_info, get_ota_train_info, hotel_search_recommand,
attractions_search_recommend, flight_search_recommend,
train_ticket_search, create_hotel_order, create_attraction_order,
create_flight_order, create_train_order, pay_hotel_order,
pay_attraction_order, pay_flight_order, pay_train_order,
search_hotel_order, search_attraction_order,
search_flight_order, search_train_order, get_hotel_order_detail,
get_attraction_order_detail, get_flight_order_detail,
get_train_order_detail, modify_train_order,
modify_flight_order, cancel_hotel_order, cancel_attraction_order,
cancel_flight_order, cancel_train_order
```

**Others (8):**
```
longitude_latitude_to_distance, address_to_longitude_latitude,
get_date_holiday_info, get_holiday_date, weather, get_nearby,
get_user_all_orders, get_user_historical_behaviors
```

## C  MODELS UNDER EVALUATION

We evaluate the following state-of-the-art language models: OpenAI GPT series (GPT-4.1, GPT-5), OpenAI o1 series (o3, o4-mini), Anthropic Claude series (Claude-4-Sonnet, Claude-4.1-Opus), Google Gemini series (Gemini-2.5-Flash, Gemini-2.5-Pro) by Anil et al. (2023), DeepSeek series (DeepSeek-V3-0324, DeepSeek-R1-0528, DeepSeek-V3.1) by DeepSeek-AI et al. (2024; 2025), Qwen3 series (Qwen3-32B, Qwen3-235B-A22B-2507) by Yang et al. (2025), and other recent language models including Kimi-K2 (Bai et al., 2025), Seed-1.6, GLM-4.5 (Zeng et al., 2025), LongCat-Flash (Li et al., 2025; Gui et al., 2025), etc. We exclude small models ($< 32B$ parameters) due to the difficulty of our benchmark. For thinking models, we follow official guidelines to enable high reasoning efforts. DeepSeek-V3.1 only supports tool calling in non-thinking mode.

Due to API stability concerns, we are currently unable to integrate some models. As our benchmark primarily focuses on Chinese contexts, we defer testing of models trained predominantly on English corpora, such as GPT-OSS-120B and Seed-OSS-36B. We further ensure that this focus does not introduce bias against the tested Chinese and European models, as the tasks are carefully designed to avoid culture-dependent knowledge and provide auxiliary information whenever such background might otherwise be required.

## D  COST AND LATENCY ANALYSIS

To enhance the transparency of the benchmark, we report representative cost and latency statistics computed under the evaluation setup described in Section 4.1.

**Representative Cost Statistics.**  Table 8 summarizes the per-task cost for two representative models (GPT-4.1 and Claude-4-Sonnet), alongside the reference values from $\tau^2$-Bench (Barres et al., 2025). These figures are generally comparable to those reported in related works, with slightly higher values in VitaBench due to the larger number of agent turns required for complex tasks.

To maintain feasibility, VitaBench adopts a four-run evaluation setting, which strikes a practical balance between statistical reliability and computational cost (Section 5.1). Additionally, the VitaBench framework supports multi-turn trajectory prefix reuse, enabling requests to benefit from input caching and further reducing overhead during multi-turn execution.

Table 8: Representative per-task cost of VitaBench.

| Benchmark | Agent Model | User Model | Agent Cost / Task | User Cost / Task |
|---|---|---|---|---|
| VitaBench | GPT-4.1 | GPT-4.1 | $0.160 | $0.021 |
| VitaBench | Claude-4-Sonnet | GPT-4.1 | $0.147 | $0.025 |
| $\tau^2$-Bench | GPT-4.1 | GPT-4.1 | $0.086 | $0.059 |

**Latency and Practical Considerations.** The overall time overhead is determined primarily by API latency of the model, the response length, and concurrency. In our runs, a GPT-4.1 trajectory completes on average in about 3.5 minutes.

## E  PROMPT TEMPLATES

The prompts used for agent system, user simulation, and sliding window evaluation are presented below.

---

**Agent System Prompt**

# Environment
- Current time: {time}

# Tool Usage Guidelines:
- When the user's needs require using tools to complete, first determine whether all parameter information is known. If it is known, extract the corresponding parameters, otherwise ask the user for the relevant parameter values
- When the user cannot provide relevant information, first obtain relevant information through tools
- Complete tasks based on Precondition and Postcondition

# Conversation Guidelines
- Only use information from the above context, prohibit constructing information without basis and replying to users
- Focus on completing user needs, prohibit divergent guidance to users to propose new needs
- After completing the user's task requirements, ask if there are any other needs. If the user indicates no, generate '###STOP###' mark to end the conversation

---

**User Simulation System Prompt**

# Role Setting
You are playing the role of a user interacting with an intelligent agent. Your character is described in the <persona> tag, and your task is to convey the content in <instructions> to the agent through user dialogue.

<persona>
{persona}
</persona>
<instructions>
{instructions}
</instructions>

# Conversation Style Rules:
- Generate only one line of content each time to simulate user messages
- **Use a combination of context description + need expression**, first describe the background situation, then express specific needs
- **When you need to make decisions, provide the conditions and preferences from**

**instructions, and let the agent help you choose**
- **Use expressions like "What do you think would be more suitable?", "Which one would you recommend?" to seek the agent's advice**
- **Must reflect the personality traits described in <persona>**, through language style, emotional expression, word choice, etc.

# Information Disclosure Rules:
- **Break down information from instructions into multiple independent points, mentioning them separately in different rounds**
- **Directly convey the original information content from instructions, but adjust the conversation style and expression according to the personality traits in <persona>**
- **Must ensure every detail from instructions is mentioned during the conversation**, even seemingly background information should be mentioned, as this information may affect the agent's recommendations and arrangements
- Avoid revealing all needs in the first round, let information unfold gradually

# Information Processing Rules:
- Answer the agent's questions based on <persona> and <instructions>. If there's no corresponding answer, reply that you don't remember or don't know
- When the agent asks for information, provide the answer immediately
- Don't fabricate information not provided in the instructions
- Strictly provide needs according to requirements explicitly stated in instructions, don't assume, expand, substitute, or generalize
- If the agent asks whether you need help placing an order, answer "Yes, please help me place the order"
- Maintain dependence on the agent's service, keep the conversation going until the task is completed
- When the agent tries to persuade you to change your needs, pay attention to sticking to the corresponding needs in <instructions>
- **If the agent repeats the same question you have already answered in the past 3 times, show impatience and refuse to answer the question**

# When NOT to End the Conversation:
- Before you clearly and completely express all needs and constraints
- Before the agent completes all tasks mentioned in instructions and confirms no operations are missed
- If the agent's execution results don't match your expectations or are incorrect/incomplete

# When You CAN End the Conversation:
- Only when all the above conditions are met and all tasks are correctly completed
- Or when you have clearly expressed complete needs but the system explicitly states it cannot complete due to technical limitations

---

Sliding Window Evaluator System Prompt

# System Information
{env_info}

# User Complete Instruction
{user_instruction}

# Background
- This is a conversation scenario evaluation between a user and an assistant, where the assistant can call tools to retrieve information and complete operations. Tool return results will start with "tool"
- Due to the large number of conversation turns, sliding window evaluation is used, where

each window shows 10 conversation turns with 2 overlapping turns between windows
- You are evaluating window {window_idx} (out of {total_windows} windows total)
- <window_content> contains the conversation content for the current window
- <current_rubrics> contains the current status of all evaluation rubrics (true means satisfied, false means not satisfied)

# Task
- Update the evaluation rubric status based on the conversation content in the current window
- All rubrics have an initial status of false, indicating incomplete. You can update the status to true, indicating the assistant completed the goal in this window
- You can also update true back to false, if and only if the assistant overturned a previous correct conclusion in this window
- You can refer to the "User Complete Instruction" to understand the progress of the current conversation window and avoid unnecessary modifications

# Important Notes
- All evaluations are based on whether the assistant's responses and tool call requests complete the goals in the rubrics
- Tool return results are only visible to the assistant and do not represent content recommended by the assistant to users
- For rubrics that require order generation, note that the assistant may mistakenly believe they completed the ordering operation when in fact the order was not successful
- For rubrics involving order details such as product quantity or delivery time, the original rubric requirements must be strictly met
- For rubrics involving text content matching of addresses or order notes, apply the functional equivalence principle

# Format Requirements
Your response should be a JSON object containing the following fields:
- `rubric_key`: Unique identifier for the rubric
- `rubric`: Restatement of the rubric
- `justification`: Explanation of status changes
- `meetExpectation`: Updated status (true or false)

# Example Input Structure:
<window_content>xxx</window_content>
<current_rubrics>xxx</current_rubrics>

# Example Response Structure:

```
[
  {
    "rubric_key": "overall_rubric_0",
    "rubric": "<restate the rubric>",
    "justification": "<brief explanation>",
    "meetExpectation": <true or false>
  },
  ...
]
```

## F  AN EXAMPLE TRAJECTORY

This section presents a complete example trajectory from VitaBench to illustrate the complexity and multi-faceted nature of our tasks. The example demonstrates a cross-scenario task that spans multiple domains (OTA for hotel booking, delivery for appliance purchase, and in-store for restaurant

reservation), requiring the agent to coordinate across different tools while managing dynamic user interactions.

The trajectory showcases several key characteristics of VitaBench:

- **Complex user profile**: The user has specific constraints (dietary restrictions, personality traits) that influence task execution

- **Multi-domain coordination**: The task requires booking a hotel, purchasing an appliance, and making a restaurant reservation

- **Implicit constraints**: Requirements like "good soundproofing" and "quiet washing machine" require inference and proactive clarification

- **Conditional logic**: The restaurant reservation depends on availability, with a fallback option (hand care treatment)

- **Spatial reasoning**: Distance calculations between hotel, dormitory, and hospital locations

The agent must navigate 66 available tools, manage incomplete information through multi-turn dialogue, and satisfy multiple constraints simultaneously. This example illustrates why even state-of-the-art models struggle with VitaBench's realistic complexity.

USER PROFILE

| System | |
|---|---|
| **User ID:** | U797215 |
| **Profession:** | Nurse |
| **Gender:** | Female |
| **Age Range:** | 25-30 |
| **Residence:** | Shijiazhuang |
| **Home Address:** | Room 502, Unit 2, Building 3, Dongfang Mingzhu Community, at the intersection of Tiyu South Street and Jianshe South Street, Yuhua District, Shijiazhuang |
| **Work Address:** | Hebei Provincial People's Hospital, No. 215 Heping West Road, Xinhua District, Shijiazhuang |
| **Dietary Restrictions:** | Avoid caffeine, avoid raw and cold foods |
| **Relationship Status:** | In a relationship |
| **Personality:** | Impatient in doing things, often urges others, lacks patience for waiting processes |
| ... | |

INSTRUCTION

> **Instruction**
>
> You've just moved to Tangshan People's Hospital for work today, with a single dormitory near the hospital. You moved in today, only to discover that both the washing machine and the drain pipe are broken. The maintenance workers won't come until the day after tomorrow, so you can only find a hotel near the hospital to make do for two nights, within 1 kilometer, just the cheapest room type. But you have sensitive nerves and need a hotel with good soundproofing.
>
> You also want to buy a new washing machine to be delivered to the dormitory, and you must choose a quiet and energy-efficient model from a major brand - Midea or Haier will do. You also need to check how far the hotel is from the dormitory to avoid inconvenient commute.
>
> To avoid affecting your mood, you want to find a Tangshan time-honored restaurant with a rating above 4.8 to dine at tonight. If reservations are possible, book a table for seven o'clock. If not, then schedule a hand care treatment instead - you really don't want to wait in line.

CONVERSATION TRAJECTORY

---

**Assistant**

[1] Hello, what service do you need?

---

**User**

[2] I just moved to Tangshan People's Hospital for work today, with a single dormitory near the hospital. As soon as I moved in, I discovered that both the washing machine and drain pipe are broken, and the maintenance workers won't come until the day after tomorrow during the day. Now I urgently need to find a hotel near the hospital for two nights, preferably within 1 kilometer walking distance. The cheapest room type is fine, but I have sensitive nerves and the soundproofing must be good. Can you help me recommend some? Which hotel would be more suitable?

---

**Assistant**

[3] *Tool Call:*
`address_to_longitude_latitude`
**Arguments:** {"address": "Tangshan People's Hospital"}

---

**Tool Call/Response**

[4] *Tool Response:*
`["118.183502", "39.630241"]`

---

**Assistant**

[5] *Tool Call:*
`get_nearby`
**Arguments:** {"latitude": 39.630241, "longitude": 118.183502, "range": 1000}

---

**Tool Call/Response**

[6] *Tool Response - Nearby Services:*

- **Stores:**
  - Little Swan Washing Machine Store (Score: 4.6, Tags: washing machines, appliances)
  - Gome Electrical Appliances Tangshan Lunan Store (Score: 4.5, Tags: appliances, washing machines, delivery)
  - ...
- **Restaurants:**
  - Tangshan Old Restaurant (Score: 4.9, Tags: time-honored brand, specialty dishes)
  - Tangshan Centennial Restaurant (Score: 4.7, Tags: traditional cuisine, specialty dishes)
  - Fumanlou Tangshan Branch (Score: 5.0, Tags: Beijing time-honored brand, traditional cuisine)
  - ...
- **Hotels:**
  - Home Inn (Tangshan People's Hospital Branch) - Score: 4.5, 2-star, Tags: wifi, good soundproofing, budget, good value
  - Ji Hotel (Tangshan City Center) - Score: 4.6, 3-star, Tags: wifi, good soundproofing, breakfast, comfortable
  - Jinjiang Inn (Tangshan Municipal Government) - Score: 4.2, 2-star, Tags: wifi, budget, busy area, lively
  - ...
- **Other Services:**
  - Beautiful Fragrance Hand Care Center (Score: 4.8, Tags: hand care, manicure, SPA)
  - ...

**Assistant**

**[7]** *Tool Call:*
`longitude_latitude_to_distance`
**Arguments:** {"latitude1": 39.630241, "latitude2": 39.631245, "longitude1": 118.183502, "longitude2": 118.184526}

**Tool Call/Response**

**[8]** *Tool Response:*
"142.0"

**Assistant**

**[9]** *Tool Call:*
`longitude_latitude_to_distance`
**Arguments:** {"latitude1": 39.630241, "latitude2": 39.629871, "longitude1": 118.183502, "longitude2": 118.179563}

**Tool Call/Response**

**[10]** *Tool Response:*
"340.0"

**Assistant**

**[11]** *Tool Call:*
`longitude_latitude_to_distance`
**Arguments:** {"latitude1": 39.630241, "latitude2": 39.636721, "longitude1": 118.183502, "longitude2": 118.180967}

**Tool Call/Response**

**[12]** *Tool Response:*
"753.0"

**Assistant**

**[13]** *Tool Call:*
`get_ota_hotel_info`
**Arguments:** {"hotel_id": "S17550802119759684_H00001"}

**Tool Call/Response**

**[14]** *Tool Response - Home Inn Details:* **Hotel:** Home Inn (Tangshan People's Hospital Branch)
**Score:** 4.5, **Star Rating:** 2-star
**Tags:** wifi, good soundproofing, budget, good value
**Available Products:**

- Standard Single Room (Oct 7): 168.0 yuan
- Standard Single Room (Oct 8): 168.0 yuan
- Standard Single Room (Oct 9): 168.0 yuan
- Business King Room (Oct 7): 228.0 yuan
- Business King Room (Oct 8): 228.0 yuan
- Standard Twin Room (Oct 7): 198.0 yuan
- Standard Twin Room (Oct 8): 198.0 yuan
- Deluxe Suite (Oct 7): 328.0 yuan
- ...

**Assistant**

**[15]** *Tool Call:*
`get_ota_hotel_info`
**Arguments:** {"hotel_id": "S17550802119759684_H00003"}

**Tool Call/Response**

**[16]** *Tool Response - Ji Hotel Details:* **Hotel:** Ji Hotel (Tangshan City Center)
**Score:** 4.6, **Star Rating:** 3-star
**Tags:** wifi, good soundproofing, breakfast, comfortable
**Available Products:**

- Standard Single Room (Oct 7): 198.0 yuan
- Standard Single Room (Oct 8): 198.0 yuan
- Business King Room (Oct 7): 268.0 yuan
- Comfort Twin Room (Oct 7): 238.0 yuan
- ...

**Assistant**

**[17]** *Tool Call:*
`get_ota_hotel_info`
**Arguments:** {"hotel_id": "S17550802119759684_H00004"}

**Tool Call/Response**

**[18]** *Tool Response - Jinjiang Inn Details:* **Hotel:** Jinjiang Inn (Tangshan Municipal Government)
**Score:** 4.2, **Star Rating:** 2-star
**Tags:** wifi, budget, busy area, lively
**Available Products:**

- Standard Single Room (Oct 7): 168.0 yuan
- Standard Single Room (Oct 8): 168.0 yuan
- Business Single Room (Oct 7): 198.0 yuan
- Business Single Room (Oct 8): 198.0 yuan
- Deluxe Single Room (Oct 7): 238.0 yuan
- ...

**Assistant**

**[19]** *Comprehensive Hotel Analysis & Recommendation:*
I've analyzed hotels within 1km with good soundproofing for your two-night stay:
**TOP RECOMMENDATION: Home Inn (Tangshan People's Hospital Branch)**

- **Distance:** 142m (2-3 min walk) - Closest to hospital
- **Rating:** 4.5 stars, budget hotel (2-star)
- **Key Features:** Good soundproofing, good value
- **Price:** Standard Single Room 168 yuan/night
- **Two-night total:** 336 yuan (Oct 7-8)
- **Why best:** Perfect combination of proximity, price, and soundproofing

**Alternative: Ji Hotel (Tangshan City Center)**

- **Distance:** 340m
- **Rating:** 4.6 stars, mid-range (3-star)
- **Features:** Good soundproofing, breakfast, comfortable
- **Price:** 198 yuan/night (396 yuan total)
- **Trade-off:** Higher comfort but 60 yuan more expensive

**Not Recommended: Jinjiang Inn**

- **Distance:** 753m - Farthest away
- **Tags:** "Busy area, lively" - Not suitable for quiet rest

> **Booking Recommendation:** Home Inn Standard Single Room for Oct 7-9 (2 nights, 336 yuan). Ready to book with your User ID. Also suggest requesting: "high floor, non-street-facing, away from elevators" for maximum quiet.
> Shall I proceed with booking Home Inn, or do you prefer Ji Hotel?

**User**

[20] This is quite urgent and really don't want to delay anymore! You said Home Inn is only 142 meters from the hospital, has good soundproofing, and the lowest price - it sounds most suitable. My user ID is U797215. Please help me quickly place an order for Home Inn Standard Single Room for two consecutive nights, check-in from October 7th to October 9th. After placing the order, remember to send me the order information so I can confirm payment quickly.

**Assistant**

[21] *Tool Calls (2):*
**Tool 1:** `create_hotel_order`
**Arguments:** {"hotel_id": "S17550802119759684_H00001", "product_id": "S17550802119759684_P00001", "user_id": "U797215"}

**Tool 2:** `create_hotel_order`
**Arguments:** {"hotel_id": "S17550802119759684_H00001", "product_id": "S17550802119759684_P00002", "user_id": "U797215"}

**Tool Call/Response**

[22] *Tool Response - Order 1 Created:*
**Order ID:** OO603df14f67
**Order Type:** hotel
**User ID:** U797215
**Hotel ID:** S17550802119759684_H00001
**Total Price:** 168.0 yuan
**Create Time:** 2025-10-07 16:30:00
**Status:** unpaid
**Product:** Standard Single Room, 2025-10-07, 168.0 yuan

**Tool Call/Response**

[23] *Tool Response - Order 2 Created:*
**Order ID:** OO31779a5b7f
**Order Type:** hotel
**User ID:** U797215
**Hotel ID:** S17550802119759684_H00001
**Total Price:** 168.0 yuan
**Create Time:** 2025-10-07 16:30:00
**Status:** unpaid
**Product:** Standard Single Room, 2025-10-08, 168.0 yuan

**Assistant**

[24] **Hotel Orders Created Successfully!**
I've created two hotel orders for your consecutive nights:
**Order 1 (October 7th):**

- Order ID: OO603df14f67
- Hotel: Home Inn (Tangshan People's Hospital Branch)
- Room: Standard Single Room
- Date: 2025-10-07
- Price: 168 yuan
- Status: Awaiting payment

**Order 2 (October 8th):**

- Order ID: OO31779a5b7f
- Hotel: Home Inn (Tangshan People's Hospital Branch)
- Room: Standard Single Room
- Date: 2025-10-08
- Price: 168 yuan
- Status: Awaiting payment

**Total Cost:** 336 yuan for two nights
Ready to process payment immediately! Please confirm and I'll handle both payments. Upon check-in, request: "high floor, non-street-facing, away from elevators and equipment rooms" for optimal quiet environment.
Shall I proceed with payment now?

---

**User**

**[25]** Yes, please help me pay for two nights quickly! I really don't want to waste any more time, please handle it fast and let me know when the payment is completed.

---

**Assistant**

**[26]** *Tool Calls (2):*
**Tool 1:** `pay_hotel_order`
**Arguments:** {"order_id": "OO603df14f67"}

**Tool 2:** `pay_hotel_order`
**Arguments:** {"order_id": "OO31779a5b7f"}

---

**Tool Call/Response**

**[27]** *Tool Response:*
Payment successful

---

**Tool Call/Response**

**[28]** *Tool Response:*
Payment successful

---

**Assistant**

**[29] Payment Completed Successfully!**
Both hotel orders have been paid:
**Order 1:** OO603df14f67 (2025-10-07 Standard Single Room) — **Payment successful**
**Order 2:** OO31779a5b7f (2025-10-08 Standard Single Room) — **Payment successful**
**Total Paid:** 336 yuan
**Check-in Ready:** You can now go directly to Home Inn (Tangshan People's Hospital Branch)
**Check-in Tips:**

- Request: "High floor, non-street-facing, away from elevators and equipment rooms"
- Ask for earplugs if available
- Mention you need a quiet room for rest

Wishing you a peaceful stay! Need any other assistance (directions, invoice, etc.)?

---

**User**

**[30]** Thank you, I received the payment information. Now I still need to buy a new washing machine to be delivered to my dormitory. It must be a quiet and energy-efficient model, and definitely from a major brand - Midea or Haier will do. Can you help me choose? Which model is more suitable for my current situation? I don't want to delay anymore, please help me recommend quickly!

---

**Assistant**

**[31]** *Tool Call:*
`longitude_latitude_to_distance`
**Arguments:** {"latitude1": 39.630241, "latitude2": 39.6287, "longitude1": 118.183502, "longitude2": 118.1823}

---

**Tool Call/Response**

**[32]** *Tool Response:*
"200.0"

---

**Assistant**

**[33]** *Tool Call:*
`longitude_latitude_to_distance`
**Arguments:** {"latitude1": 39.630241, "latitude2": 39.629543, "longitude1": 118.183502, "longitude2": 118.187652}

---

**Tool Call/Response**

**[34]** *Tool Response:*
"364.0"

---

**Assistant**

**[35]** *Washing Machine Consultation & Purchase Plan:*
Understood! You need "quiet + energy-efficient + major brand" with fastest delivery. Here's the optimized selection plan:
**Recommended Specifications:**

- **Type:** Front-loading (drum) 8kg
- **Motor:** BLDC direct-drive inverter motor
- **Energy:** Level 1 efficiency rating
- **Noise:** Washing ≤50dB, Spinning ≤72dB (or lower)
- **Functions:** Night wash/delayed wash, drum self-cleaning
- **Extras:** Shock-absorbing pads, motor warranty ≥10 years

**Nearby Purchase Options:**

- **Gome Electrical Appliances (Tangshan Lunan Store):** 364m away
- **Little Swan Washing Machine Store:** 200m away
- Both support delivery and installation services

**Brand Strategy:** Either Haier or Midea is fine - choose whichever meets specs and has lower noise ratings. When similar noise levels, prioritize lower price.
**Quick Selection Script for Store:** *"I want Haier/Midea, front-loading 8kg, BLDC direct-drive, Level 1 energy efficiency, washing ≤50dB/spinning ≤72dB, with night wash and drum self-cleaning. Order today, delivery/installation October 9th daytime to dormitory near Tangshan People's Hospital with shock-absorbing pads and on-site leveling."*
**Installation Plan:** October 9th daytime delivery with professional installation, leveling, and noise testing.
Ready to help you contact Gome store with complete pre-booking message including your details for immediate confirmation. Need your contact info to generate the ready-to-send message?

---

**User**

**[36]** I'm really anxious right now, please help me contact Gome Electrical Appliances (Tangshan Lunan Store)! Just use the parameters you mentioned - either Haier or Midea is fine, whoever has better noise reduction choose that one. Make sure to schedule October 9th daytime delivery and installation to my dormitory, the address is single dormitory near Tangshan People's Hospital. Please arrange it quickly and don't keep me waiting too long!

**Assistant**

[37] *Contact Preparation for Gome Store:*
I'll prepare everything for immediate contact! I cannot directly call the store, but I'll create a complete pre-booking message for you to copy and send directly to Gome Electrical Appliances.
**Store Information:**

- **Name:** Gome Electrical Appliances (Tangshan Lunan Store)
- **Address:** No. 120 Xinhua East Road, Lunan District, Tangshan
- **Distance:** 364m from hospital (5-min walk)
- **Services:** Delivery and installation available

**Pre-booking Message Template:** *Ready to customize with your contact details for immediate sending*
**30-Second Phone Script:** *Prepared for direct reading if you prefer calling*
**To Complete Setup:** I need:

1. Your name
2. Mobile number
3. Detailed dormitory address (building/room/access info)

Once you provide these 3 items, I'll generate the complete message for one-click copying and immediate store contact!
Ready to provide your details for instant message generation?

**User**

[38] I need to handle this right now, no delays! Recipient name is Wang Jing, mobile number is 13812345678, dormitory address is near Tangshan People's Hospital, specifically Room 502, Unit 2, Building 3, Dongfang Mingzhu Community, access is by card swipe for unit door. Please help me generate the complete pre-booking text quickly, I want to contact the store immediately!

**Assistant**

[39] **Complete Pre-booking Message Generated - Ready to Send:**
**Copy and Send to Gome Store:**

> Hello Gome Electrical Appliances (Tangshan Lunan Store), I want to purchase a drum washing machine and schedule delivery/installation. Requirements as follows:
> **Brand:** Haier or Midea, choose whichever is quieter
> **Specifications:** 8kg drum, BLDC direct-drive inverter motor, Level 1 energy efficiency
> **Noise indicators:** Washing $\leq$50dB, spinning $\leq$72dB (or lower)
> **Functions:** Night wash/delayed wash, drum self-cleaning
> **Installation:** October 9th daytime (9:00-17:00) delivery/installation to "Room 502, Unit 2, Building 3, Dongfang Mingzhu Community, near Tangshan People's Hospital," access: card swipe for unit door. Bring shock-absorbing pads and level on-site, test run to verify noise and stability.
> **Old machine:** Broken, please quote removal/recycling if available
> **Selection:** Prioritize better noise reduction; if similar noise, choose more affordable option.
> **Contact:** Wang Jing, 13812345678
> **Address:** Room 502, Unit 2, Building 3, Dongfang Mingzhu Community
> **Time:** October 9th 9:00-17:00 (please call before delivery)

**Alternative Phone Script:** *"Hello, I want to buy a quiet washing machine, Haier or Midea, 8kg drum, BLDC direct-drive, Level 1 energy efficiency, under 50dB washing/72dB spinning, with night wash and self-cleaning. October 9th delivery to Room 502, Unit 2, Building 3, Dongfang Mingzhu Community near Tangshan People's Hospital. Wang Jing, 13812345678. Prioritize noise reduction, quote old machine removal. Please confirm model and price ASAP."*
Ready for immediate store contact! I'll help you evaluate their response and choose the quietest model when they reply.

**User**

**[40]** Please help me monitor the store's reply first. I don't want to compare models slowly myself. After they give specific models and prices, you can directly help me choose the one with the best noise reduction effect. Don't make me worry about it again! Also, I want to go to a Tangshan local time-honored restaurant with a rating above 4.8 for dinner tonight, preferably book a table for seven o'clock in advance so I don't have to queue. Please help me check quickly, which restaurant is most suitable?

**Assistant**

**[41]** *Tool Call:*
`longitude_latitude_to_distance`
**Arguments:** {"latitude1": 39.630241, "latitude2": 39.629876, "longitude1": 118.183502, "longitude2": 118.186231}

**Tool Call/Response**

**[42]** *Tool Response:*
"237.0"

**Assistant**

**[43]** *Tool Call:*
`longitude_latitude_to_distance`
**Arguments:** {"latitude1": 39.630241, "latitude2": 39.630125, "longitude1": 118.183502, "longitude2": 118.180254}

**Tool Call/Response**

**[44]** *Tool Response:*
"278.0"

**Assistant**

**[45]** *Parallel Task Management:*
Perfect! I'll handle both tasks simultaneously:
**1. Washing Machine Follow-up Strategy:**

- I cannot directly receive store messages
- Please forward Gome store's reply with: model/noise dB/energy efficiency/motor type-/price/stock/delivery time
- I'll rank by: **Noise priority** (lower washing/spinning dB) > BLDC direct-drive > Energy efficiency > Price
- Will provide direct "buy this one" decision with reasoning
- Ensure October 9th daytime delivery/installation to your dormitory

**2. Restaurant Recommendation & Reservation (19:00):**
**Top Choice: Tangshan Old Restaurant**

- Rating: 4.9 (>4.8 requirement)
- Type: Local time-honored brand
- Distance: 237m from hospital
- Reservation: Supports free booking

**Alternative: Fumanlou (Tangshan Branch)**

- Rating: 5.0 (>4.8 requirement)
- Type: Beijing time-honored brand
- Distance: 278m from hospital
- Reservation: Supports free booking

**Recommendation:** Tangshan Old Restaurant - local time-honored, highest rating above 4.8, closest distance.

Please confirm number of diners for 19:00 reservation. If no reply, I'll default to 1 person and book immediately with note for "quiet seating, away from entrance and cashier."

**User**

**[46]** Just me dining alone, please quickly help me book tonight's seven o'clock table at Tangshan Old Restaurant, remember to note that I want a quiet location, don't keep me waiting too long! Let me know the result quickly after booking.

**Assistant**

**[47]** *Tool Call:*
`instore_book`
**Arguments:** {"customer_count": 1, "shop_id": "S17550802119759684_I00006", "time": "2025-10-07 19:00:00", "user_id": "U797215"}

**Tool Call/Response**

**[48]** *Tool Response - Booking Successful:*
**Booking ID:** OI25b43c4738
**Shop ID:** S17550802119759684_I00006
**Booking Time:** 2025-10-07 19:00:00
**Customer ID:** U797215
**Customer Count:** 1
**Booking Price:** 0.0 yuan
**Status:** paid

**Assistant**

**[49] Restaurant Reservation Confirmed!**
**Booking Details:**

- **Restaurant:** Tangshan Old Restaurant (4.9 rating, time-honored brand)
- **Date & Time:** Tonight 2025-10-07 19:00
- **Party Size:** 1 person
- **Reservation ID:** OI25b43c4738
- **Fee:** 0 yuan (free reservation)
- **Special Request:** Quiet seating arranged (away from entrance, cashier, kitchen)

**Arrival Tips:**

- Arrive 5-10 minutes early
- Mention: "Reservation ID OI25b43c4738 + Wang Jing"
- If running late, call restaurant in advance

**Washing Machine Status:** Ready to evaluate Gome store's reply. Send me their model options (noise levels, prices, stock, delivery time) and I'll provide immediate "buy this one" decision with confirmation script for October 9th delivery/installation.
Need navigation to restaurant or dish recommendations? Otherwise, standing by for washing machine store response.

**User**

**[50]** No need, the restaurant booking is fine. Just help me monitor and choose the washing machine model with the best noise reduction when the store replies, I'll send it to you when they get back to me. Now that these things are arranged, I don't have any other needs for now.

**Assistant**

**[51] Task Status Summary:**
**Hotel:** Home Inn (Tangshan People's Hospital Branch) - **Payment successful**
 Two nights booked, 336 yuan paid, ready for check-in today

**Restaurant:** Tangshan Old Restaurant - **Reserved**
Tonight 19:00, 1 person, ID: OI25b43c4738, quiet seating noted
**Washing Machine:** **Awaiting store response**
Ready to evaluate models by "noise priority > BLDC > efficiency > price"
Will provide direct purchase decision for October 9th delivery/installation
**Next Steps:** Standing by for Gome store reply. Once received, I'll immediately analyze and provide final purchase recommendation.
Wishing you a pleasant dinner tonight and peaceful rest at the hotel!
**###STOP###**

