# OpenReview forum: "VitaBench: Benchmarking LLM Agents with Versatile Interactive Tasks in Real-world Applications"
_ICLR.cc/2026/Conference — ICLR 2026 Poster_

### Official Review · Reviewer_zsYD · 2025-10-29

**Soundness:** 3
**Presentation:** 3
**Contribution:** 2
**Rating:** 6
**Confidence:** 3

**Summary:**

This paper introduces VitaBench, a benchmark designed to evaluate LLM-based agents in real-life applications. VitaBench formalizes task complexity along three dimensions—reasoning, tool, and interaction—and proposes a rubric-based sliding window evaluator to assess multi-turn trajectories. With the extensive experiments conducted, the author show that even the best current models achieve only around 30% success in cross-scenario tasks, underscoring the remaining challenges in developing capable real-world LLM agents.

**Strengths:**

- This paper is well-organized and clearly written.

- This paper addresses the gap in the absence of evaluations of LLM Agents in real-world settings.

- The proposed task complexity formalization provides a reasonable way to quantify the task difficulties.

**Weaknesses:**

- Although the paper claims to benchmark “LLM-based agents” in real-world scenarios, the evaluation is actually conducted on single LLM models acting as direct function-calling agents, rather than on established agent frameworks or architectures.

- The benchmark evaluates current models as static agents without adaptation or learning. It remains unclear how VitaBench can guide or support the training of future agent models beyond static evaluation.

**Questions:**

Have the authors considered evaluating existing agent frameworks？

---

> ### Author Response · Authors · 2025-11-17
> **Official Response from VitaBench Authors**
>
> Dear Reviewer zsYD,
>
> Thank you for your valuable insights and suggestions, and for highlighting our strengths (i.e., **the paper writing** and **the task complexity formalization**). In the following, we will carefully address your questions.
>
> ---
>
> 1. **Question about why we only evaluate foundation LLMs (direct function-calling agent).**
>    - Thank you for raising this point. Our choice to evaluate single LLM models with a minimal function-calling setting is intentional. The benchmark is designed to measure the **intrinsic agentic and tool-use capabilities of recent foundation LLMs**, rather than the performance of complex, pre-engineered agent frameworks. So we adopt the simplest and most commonly used function-calling pattern (**minimal design**) to ensure clear model-level comparability.
>    - In addition, our tasks require the LLM to act as an assistant interacting across diverse domains, which makes heavy or task-specific frameworks or architectures difficult to apply; **this is also the case for other TauBench-like settings** [1]. It differs withcode-centric benchmarks such as SWE-Bench [2], where a more elaborate framework (e.g., OpenHands [3]) that can substantially benefit performance.
>    - We acknowledge that more complex agent designs—such as domain-specific prompting, memory modules, or test-time scaling—can indeed improve results in VitaBench. Our Pass@4 experiments already show that even simple majority voting may yield gains. While these techniques are valuable in practice, we avoid integrating them into the benchmark **to maintain clear, model-level comparability**. We appreciate your understanding!
>    - We also thank you for your feedback on terminology. In the updated version, we will replace "LLM-based agents" with more precise terms such as **"agentic models" or "foundation models with agentic abilities"** to prevent any potential misunderstanding.
> 2. **Question about the support for adaptive or trainable agent models.**
>    - As mentioned before, our focus is to evaluate the capabilities of foundation LLMs rather than evolving agent systems. Nevertheless, even as a static evaluation, VitaBench remains valuable for foundation model development. The systematic formalization of Agentic Task Complexity **provides clearer insights into model weaknesses and supports the creation of complexity-specific datasets** if needed, which can in turn guide model improvement.
>    - In addition, VitaBench offers a complete framework, including a runnable environment, an interactive user simulator, and a reliable evaluator. **With this infrastructure, researchers can construct new tasks, generate fine-tuning samples, or conduct RL training directly on the tasks**. Similar efforts have been explored in settings like Tau-Bench [4, 5], and VitaBench can support comparable workflows.
>
> ---
>
> **Reference**
>
> [1] Tau2-Bench: Evaluating Conversational Agents in a Dual-Control Environment, arXiv: 2506.07982
>
> [2] SWE-bench: Can Language Models Resolve Real-World GitHub Issues? ICLR 2024
>
> [3] OpenHands: An Open Platform for AI Software Developers as Generalist Agents, ICLR 2025
>
> [4] MUA-RL: Multi-turn User-interacting Agent Reinforcement Learning for Agentic Tool Use, arXiv:2508.18669
>
> [5] Process-Supervised Reinforcement Learning for Interactive Multimodal Tool-Use Agents, arXiv:2509.14480

---

> > ### Comment · Reviewer_zsYD · 2025-11-27
> >
> > Thanks for the response. I will maintain the positive rating.

---

> > > ### Author Response · Authors · 2025-11-27
> > > **Thank You for Your Follow-Up and Consideration**
> > >
> > > Dear Reviewer zsYD,
> > >
> > > Thank you very much for your follow-up and consideration. We are glad to hear that our responses have addressed your concerns.
> > >
> > > Once again, we sincerely appreciate the time and effort you spent reviewing our paper, and we are grateful for your positive assessment of our work.
> > >
> > > Best regards,
> > >
> > > Authors of VitaBench

---

### Official Review · Reviewer_boYB · 2025-10-30

**Soundness:** 3
**Presentation:** 3
**Contribution:** 2
**Rating:** 6
**Confidence:** 3

**Summary:**

This paper presents VitaBench, a benchmark designed to evaluate agents across three interactive daily-life scenarios: food delivery, in-store consumption, and online travel services. The benchmark comprises 100 cross-scenario tasks, 300 single-scenario tasks, and 66 tools. Agent performance is assessed using a rubric-based sliding window evaluator to handle long-horizon trajectories. Experiments on multiple LLMs demonstrate the difficulty of the benchmark, as the best-performing model achieves only a 30% success rate in the cross-scenario setting.

**Strengths:**

1.	The paper is well-written and clearly describes the benchmark and evaluation results.
2.	The benchmark focuses on realistic, real-world scenarios, making it valuable for guiding real agent development. Its components, including the user simulator, evaluator, and hyperparameter settings, are shown to be robust in the reliability analysis.
3.	The paper provides thorough analysis of the evaluation results, from aspects of reasoning complexity, task complexity, and interactive complexity.

**Weaknesses:**

1.	Although the benchmark is well designed, there already exist many benchmarks addressing similar tasks. It is unclear how much new insight VitaBench contributes beyond existing efforts.
2.	The sliding window evaluator is intended to manage long trajectories, but it raises the question of whether evaluation should occur over the entire trajectory. An alternative approach could be to extract key outcomes (e.g., booking results) and apply the rubrics to those results directly. Additionally, the permanent satisfaction criterion across windows may fail when an agent meets a rubric early but changes later.

**Questions:**

1.	Could you provide more details about the tools? Specifically, what are the 66 tools, and are their returned values (e.g., store_info) real or simulated?
2.	Do variations in simulated user attributes or communication styles influence the results differently?
3.	What does the “score” in Table 4 represent?

---

> ### Author Response · Authors · 2025-11-17
> **Official Response from VitaBench Authors [1/2]**
>
> Dear Reviewer boYB,
>
> Thank you for your valuable insights and suggestions, and for highlighting our strengths (i.e., **the paper clarity**, **the real-world scenarios**, and **the thorough analysis**). In the following, we carefully address your questions.
>
> ---
>
> 1. **Question about the novelty of VitaBench beyond existing benchmarks.**
>    - Our primary novelty lies in the systematic formalization of **Agentic Task Complexity**, which is not involved in existing benchmarks. This theoretical foundation enables us to quantify and improve three key dimensions—reasoning, tool use, and interaction complexity, as detailed in Table 1 of our paper. These enhancements allow for **a more comprehensive evaluation of current models**, setting VitaBench apart from other benchmarks.
>    - Furthermore, this framework guides model improvement by **providing clearer insights into model weaknesses and supporting the creation of complexity-specific datasets**, if needed. For example, in RL training, models can more effectively explore and exploit their performance in environments at their performance boundaries.
>    - VitaBench also introduces **an extensible environment and tool set**, supported by a framework that eliminates domain-specific policies and enables cross-scenario composition. This design ensures the benchmark’s scalability, offering significant potential for **environment scaling** [1] in the future.
>
> 2. **Question about the sliding window evaluator and its effectiveness over entire trajectories.**
>    - As you pointed out, comparing the final state of the database is a common approach in existing works (e.g., in TauBench [2]). However, we argue that this method does **not fully capture the scope of tasks we aim to evaluate** (as discussed in Section 3.3 of the paper), such as those involving recommendations or planning behaviors, which may not result in a final state change. Additionally, it does not provide supervision for intermediate transitions.
>    - Regarding your concern about the satisfaction criterion across windows, we believe there may be a misunderstanding. In our setup, **the rubric state is flexible and can be adjusted as needed**. For example, if an agent initially takes an action (e.g., placing an order) and the rubric is marked as 1, but later cancels the order, the rubric state can be reset to 0. The evaluator tracks this cross-window information, ensuring that any changes are reflected accurately.
>
> ---
>
> **Reference**
>
> [1] Towards General Agentic Intelligence via Environment Scaling, arXiv: 2509.13311
>
> [2] Tau-bench: A Benchmark for Tool-Agent-User Interaction in Real-World Domains, ICLR 2025

---

> ### Author Response · Authors · 2025-11-17
> **Official Response from VitaBench Authors [2/2]**
>
> 3. **Question about the tools used in VitaBench.**
>
>    - Sure, the tools in VitaBench are abstractions of the APIs required in real-world scenarios. Each tool is implemented as a Python function, and **all returned values are drawn from a database in the equipped sandbox, ensuring consistency and determinism across executions**. Below is the complete list of all 66 tools:
>      - Delivery (12): `delivery_distance_to_time, get_delivery_store_info, get_delivery_product_info, delivery_store_search_recommand, delivery_product_search_recommand, create_delivery_order, pay_delivery_order, get_delivery_order_status, cancel_delivery_order, modify_delivery_order, search_delivery_orders, get_delivery_order_detail`
>      - In-store (16): `instore_shop_search_recommend, instore_product_search_recommend, create_instore_product_order, pay_instore_order, instore_cancel_order, instore_book, pay_instore_book, instore_cancel_book, instore_reservation, instore_modify_reservation, instore_cancel_reservation, get_instore_orders, get_instore_reservations, get_instore_books, search_instore_book, search_instore_reservation`
>      - OTA (30): `get_ota_hotel_info, get_ota_attraction_info, get_ota_flight_info, get_ota_train_info, hotel_search_recommand, attractions_search_recommend, flight_search_recommend, train_ticket_search, create_hotel_order, create_attraction_order, create_flight_order, create_train_order, pay_hotel_order, pay_attraction_order, pay_flight_order, pay_train_order, search_hotel_order, search_attraction_order, search_flight_order, search_train_order, get_hotel_order_detail, get_attraction_order_detail, get_flight_order_detail, get_train_order_detail, modify_train_order, modify_flight_order, cancel_hotel_order, cancel_attraction_order, cancel_flight_order, cancel_train_order`
>      - Others (8): `longitude_latitude_to_distance, weather, address_to_longitude_latitude, get_date_holiday_info, get_holiday_date, get_user_historical_behaviors, get_user_all_orders, get_nearby`
>
> 4. **Question about the influence of simulated user attributes on results.**
>
>    - Yes, variations in simulated user attributes do influence the results. In VitaBench, each task is assigned a persona at random, and we further conducted an additional experiment to quantify the effect. We evaluated Claude-4-Sonnet on 100 cross-scenario tasks while **fixing the user simulator to one of five personas**, and report the resulting success-rate distribution:
>
>      | Persona              | Avg@4 | Pass@4 | Pass^4 |
>      | :------------------- | :---: | :----: | :----: |
>      | Random (as in paper) | 21.3  |  49.0  |  4.0   |
>      | Impatient            | 21.5  |  48.0  |  4.0   |
>      | Anxious              | 18.5  |  41.0  |  2.0   |
>      | Scattered            | 19.3  |  47.0  |  0.0   |
>      | Dependent            | 20.6  |  45.0  |  3.0   |
>      | Cooperative          | 22.8  |  50.0  |  5.0   |
>
>    - The results indicate that certain personas lead to slightly higher performance, while **anxious or scattered** personas may reduce both performance and stability.
>
> 5. **Question about the meaning of "score" in Table 4.**
>
>    - Sorry for the unclear explanation in the paper. The "score" refers to **the evaluation score assigned to corresponding GLM-4.5 trajectories after applying the respective method**. For example, the baseline (using our rubric-based sliding window evaluator) yields a score of 20, closely matching the real performance level of GLM-4.5. In contrast, when the rubric is ablated, the score inflates to over 80.
>    - We will clarify this in the updated version of the paper for better clarity.

---

> > ### Comment · Reviewer_boYB · 2025-11-26
> >
> > Thank the authors for the rebuttal!
> >
> > For the second point regarding the sliding window evaluator, in the original paper, it is stated that "once a rubric item rj is satisfied in any window, sj is permanently marked." This differs from what the authors stated in their rebuttal.
> >
> > Nevertheless, most of my concerns have been addressed, and I will maintain my positive score.

---

> > > ### Author Response · Authors · 2025-11-26
> > > **Thank You for Your Follow-Up and Consideration**
> > >
> > > Dear Reviewer boYB,
> > >
> > > Thank you very much for your follow-up and for confirming that our responses have addressed most of your concerns.
> > >
> > > Regarding the point on the sliding-window evaluator: you are right that our original wording (“*permanently marked*”) was unclear. Our intention was to convey that the state vector persists across windows, rather than implying irreversibility. We have updated the paper to avoid this ambiguity (**highlighted in blue**):
> > >
> > > > once a rubric item $r_j$ is satisfied in any window, **$s_j$ is marked as $1$, and if a previously satisfied criterion is later negated, the corresponding state will be reset to $0$ accordingly.**
> > >
> > > We sincerely appreciate your careful reading and constructive feedback, which has helped us improve the clarity of the paper! We also appreciate your positive rating, and if this clarifications is satisfactory, we would be grateful if you could consider increasing your confidence level as well.
> > >
> > > Best regards,
> > >
> > > Authors of VitaBench

---

> > > > ### Comment · Reviewer_boYB · 2025-11-26
> > > >
> > > > I have updated my confidence.

---

> > > > > ### Author Response · Authors · 2025-11-26
> > > > > **Thank you!**
> > > > >
> > > > > The authors thank you for your kind feedback!

---

### Official Review · Reviewer_Bj6N · 2025-10-30

**Soundness:** 2
**Presentation:** 3
**Contribution:** 2
**Rating:** 4
**Confidence:** 4

**Summary:**

This paper introduces VitaBench, a new benchmark designed to evaluate LLM-based agents on complex, interactive tasks grounded in real-world, "life-serving" applications. The benchmark spans three domains (food delivery, in-store consumption, online travel) and comprises 66 tools, 100 cross-scenario tasks, and 300 single-scenario tasks. The authors motivate their work by arguing that existing benchmarks lack the necessary complexity in terms of information volume, tool interdependencies, and dynamic user interaction.

**Strengths:**

1. The paper correctly identifies a critical need for more challenging and realistic agent benchmarks. VitaBench represents a significant step up in scale and complexity compared to prior work, pushing the evaluation frontier toward long-horizon. The ambition to create a "life-serving simulation environment" is good and moves the field in the right direction.

2. The proposed rubric-based sliding window evaluator is a  practical solution to the problem of evaluating long-form agent trajectories with LLMs.

**Weaknesses:**

1. The benchmark's entire pipeline is built on LLMs. The agent is an LLM, the user simulator is an LLM (gpt-4.1), and the final evaluator is another LLM (claude-3.7-sonnet). While the authors conduct reliability studies (Section 5.1), this "LLM-evaluating-LLM-interacting-with-LLM" setup raises concerns about potential biases and circularity. The benchmark may inadvertently measure how well one model's behavior and style align with another's, rather than objective task success. For example, the user simulator's "cooperative" nature might favor agents with similar inherent tendencies, and the evaluator might be more lenient towards reasoning patterns similar to its own. This methodological concern is significant and undermines the objective grounding of the benchmark's results.

2. The simulator is prompted to faithfully convey all the points from a pre-written, detailed instruction set (page 13, "Must ensure every detail from instructions is mentioned"). Real users are not like this. They forget their own constraints, provide contradictory information ("I want the cheapest option... no, not that one, it looks bad"), and change their minds midway through a task. A key skill for a real agent is navigating this human inconsistency, which VitaBench's user doesn't seem to model. The paper notes that user behavior like "impatience" is explicitly prompted (e.g., "If the agent repeats the same question... show impatience," line 719). This is a scripted reaction. A real impatient user might interrupt, abandon the conversation, or start making demands outside the original scope. Moreover, The agent can make as many tool calls as it wants without penalty. In reality, API calls can have monetary costs, and excessive interaction turns frustrate the user and increase the probability of abandonment. **I think the benchmark tests an agent's ability to follow a complex conversational script and using tools rather than handle truly emergent, unpredictable human behavior thus not a realisttic benchmark.**

**Questions:**

The benchmark is explicitly focused on "Chinese contexts" (Appendix B, line 646), while many of the top-performing models evaluated (like the GPT and Claude series) are developed by Western companies and primarily trained on English-language and Western-centric data.Many reasoning tasks are steeped in cultural common sense. For example, the user request in Figure 3 mentions booking a dinner for a "10th anniversary celebration" near the "Huangpu River." While anniversary is a universal concept, the specific expectations around such an event, typical restaurant choices, or etiquette might differ. A model trained on Western data might make different assumptions than one with deeper knowledge of modern Chinese urban culture. The example trajectory's mention of a "Tangshan time-honored restaurant" is a perfect example of a culturally specific concept (老字号, lǎo zì hào) that an English-centric model might only understand at a surface level. I was wondering if author considering this.

---

> ### Author Response · Authors · 2025-11-17
> **Official Response from VitaBench Authors**
>
> Dear Reviewer Bj6N,
>
> Thank you for your valuable insights and suggestions, and for highlighting our strengths (i.e., **the benchmark’s realism and complexity**, and **the evaluator**). In the following, we carefully address your questions.
>
> ---
>
> 1. **Question about LLM-only evaluation pipeline and potential biases.**
>
>    - We fully acknowledge the risk of circularity in LLM–LLM interactions. However, introducing real human users in the evaluation at scale is infeasible, so **an LLM-based user simulator is unavoidable**. To remain consistent with prior work (e.g., Tau2-Bench [1]) and ensure comparability, we use GPT-4.1 as the simulator. We clearly specify and tune its behavior, and the human evaluation in Section 5.1 finds **no explicit cooperative tendencies**.
>
>    - We also conduct controlled cross-model experiments to examine potential implicit cooperation (Table below). When replacing the user simulator with Claude-4-Sonnet, GPT-4.1’s performance remains nearly unchanged, whereas Claude-4-Sonnet’s performance drops slightly. This asymmetry suggests that **any implicit cooperation is minimal**.
>
>      | User Simulator  |   Agent Model   |     Evaluator     | Avg@4 | Pass@4 | Pass^4 |
>      | :-------------: | :-------------: | :---------------: | :---: | :----: | :----: |
>      |     GPT-4.1     |     GPT-4.1     | Claude-3.7-Sonnet | 13.8  |  35.0  |  0.0   |
>      |     GPT-4.1     | Claude-4-Sonnet | Claude-3.7-Sonnet | 21.3  |  49.0  |  4.0   |
>      | Claude-4-Sonnet |     GPT-4.1     | Claude-3.7-Sonnet | 13.8  |  34.0  |  0.0   |
>      | Claude-4-Sonnet | Claude-4-Sonnet | Claude-3.7-Sonnet | 19.5  |  45.0  |  5.0   |
>
>    - For the evaluator, in fact, we deliberately **design all rubrics to be objective, fine-grained and binary** (0/1), minimizing dependence on phrasing, style, or evaluator-specific preferences. The human–evaluator agreement (Section 5.1) further indicates that the evaluator captures task success reliably.
>
> 2. **Question about the realism of the user simulator and lack of emergent human behavior.**
>
>    - Actually, our first design principle for VitaBench is **evaluation stability and reliability**. Fully modeling unconstrained human behavior would introduce substantial randomness, and, when users omit key information, could even make tasks unsolvable—undermining the goal of a controlled and comparable evaluation.
>    - We also experimented with more flexible user behaviors, but ultimately opted for a stable version. Within this constraint, we incorporated as much realistic complexity as possible. For instance, several tasks explicitly script users "changing their mind" or updating earlier intents, requiring the agent to reverse previous actions (e.g., place → cancel → re-place an order). **While these behaviors are scripted by design, they still reflect real interaction patterns while maintaining a deterministic benchmark.**
>    - The inclusion of impatience cues is also intentional. Well-aligned LLMs tend to be overly sycophantic [2]; without such signals, the simulator would become unrealistically cooperative. We therefore combine explicit instructions with persona-based prompting to better approximate natural conversational dynamics. Given the constraints of a deterministic evaluation, **this is the most effective way to balance realism and stability**.
>    - To the best of our knowledge (see Table 1 in our paper), existing agent benchmarks use even simpler user simulators. **VitaBench aims to push this frontier**, and we agree that richer, more emergent modeling of human behavior is an important direction for future work.
>
> 3. **Question about cultural specificity and concerns for English-centric models.**
>
>    - Thank you for your insightful question regarding fairness. We acknowledge that there may be limitations in the cultural specificity of our benchmark, largely due to the data sources we used. To address this, **we actively designed the benchmark to minimize potential biases**, ensuring that scenarios requiring specific cultural knowledge were avoided. When cultural context might be important (e.g., dining etiquette), we provided additional auxiliary information to help the model complete tasks without overly relying on cultural background.
>    - From another perspective, while cultural specificity is certainly present, the scale of modern LLM pretraining, particularly with massive data that include extensive Chinese corpora, has significantly **narrowed the gap between Eastern and Western models**. As reported in CulturalBench [3], *Chinese and European model providers do not exhibit a clear advantage in their respective regional cultural knowledge* (Section 4.3).
>
> ---
>
> **Reference**
>
> [1] Tau2-Bench: Evaluating Conversational Agents in a Dual-Control Environment, arXiv: 2506.07982
>
> [2] Towards Understanding Sycophancy in Language Models, ICLR 2024
>
> [3] CulturalBench: A Robust, Diverse, and Challenging Cultural Benchmark by Human-AI CulturalTeaming, ACL 2025

---

> ### Author Response · Authors · 2025-11-27
> **A Friendly Reminder from VitaBench Authors**
>
> Dear Reviewer Bj6N,
>
> We hope this message finds you well. If this email arrives outside your normal working hours, please accept our apologies for the interruption.
>
> **It has been 10 days since our last round of communication**. We just want to kindly follow up to ensure that we have addressed your concerns or any remaining questions you might have. We are still here and welcome any further discussion or feedback, as your insights are incredibly valuable to us.
>
> Thank you very much for your time and consideration.
>
> Best regards,
>
> Authors of VitaBench

---

### Official Review · Reviewer_jGAh · 2025-10-31

**Soundness:** 3
**Presentation:** 3
**Contribution:** 3
**Rating:** 6
**Confidence:** 3

**Summary:**

This paper introduces VitaBench, a real-world–oriented agent benchmark spanning three domains (delivery, in-store consumption, OTA).
The tasks require multi-turn dialogue, temporal/spatial reasoning, proactive clarification, and tool chaining. Results show strong difficulty: ~30% success on cross-scenario and <50% on single-scenario tasks with various state-of-the-art models.

**Strengths:**

1. Clear three-axis task-complexity framework (reasoning/tool/interaction) and cross-scenario composition without domain-specific policies.
2. The data are collected from authentic platform data.
3. Rubric + sliding window evaluator with reported human agreement (κ≈0.828) and ablations justifying design choices.
4. Comprehensive experiments and error taxonomy (reasoning dominates) study,

**Weaknesses:**

1. Compute/latency opacity: Many-turn trajectories and 4-run protocol imply high cost; paper lacks concrete cost accounting per task/model.
2. The main results are based on only one basic function calling agent. It is unknown how different agent design would affect the results.

**Questions:**

1. What is the cost analysis of different models?
2. Is any other agents/techniques being tested? Can agent design affect results?

---

> ### Author Response · Authors · 2025-11-17
> **Official Response from VitaBench Authors**
>
> Dear Reviewer jGAh,
>
> Thank you for your valuable insights and suggestions, and for highlighting our strengths (i.e., **the benchmark design and the experiments**). In the following, we will carefully address your questions.
>
> ---
>
> 1. **Question about the cost and latency analysis across models.**
>
>    - Thank you for pointing this out. In fact, our work is designed to **probe the upper-bound capabilities** of recent foundation LLMs under stable evaluation settings, so we did not emphasize cost-sensitive comparisons.
>
>    - To reduce opacity, we have added **cost statistics** for two representative models (GPT-4.1 and Claude-4-Sonnet; computed directly using their official pricing rules, and other models can be derived in the same manner) and compared them with the reported numbers from Tau2-Bench [1], as shown in the table below. These values are **generally comparable to those reported in similar works**, with slightly higher values due to more agent turns in VitaBench. We will include this cost analysis in the updated version of the paper.
>
>      | Benchmark      |   Agent Model   | User Model | Agent Cost / Task | User Cost / Task |
>      | :------------- | :-------------: | :--------: | :---------------: | :--------------: |
>      | VitaBench      |     GPT-4.1     |  GPT-4.1   |      \$0.160      |     \$0.021      |
>      | VitaBench      | Claude-4-Sonnet |  GPT-4.1   |      \$0.147      |     \$0.025      |
>      | Tau2-Bench [1] |     GPT-4.1     |  GPT-4.1   |      \$0.086      |     \$0.059      |
>
>    - Regarding latency, the overall time overhead is determined primarily by API latency of the model, the response length, and concurrency. In our runs, a GPT-4.1 trajectory completes on average **in about 3.5 minutes**.
>
>    - Finally, to keep the benchmark accessible we (1) conducted an experiment in Section 5.1 showing that the 4-run setting strikes a reasonable **balance between statistical precision and cost**, and (2) adopt multi-turn trajectory prefix-reuse so that LLM API requests can benefit from **input caching** [2], thereby significantly reducing the cost of multi-turn interactions
>
> 2. **Question about why we only evaluate foundation models (basic function-calling agent).**
>
>    - Thank you for raising this question. Our choice to evaluate only foundation models with a minimal function-calling setting is intentional. The benchmark is designed to measure the **intrinsic agentic and tool-use capabilities of recent foundation LLMs**, so we adopt the simplest and most commonly used function-calling pattern (**minimal design**), aiming for a leaderboard that is informative for everyday model selection and future model development.
>    - In addition, our tasks require the LLM to act as an assistant interacting across diverse domains, which makes heavy or task-specific design difficult to apply; **this is also the case for other TauBench-like settings** [1]. It differs with code-centric benchmarks such as SWE-Bench [3], where a more elaborate framework (e.g., OpenHands [4]) that can substantially benefit performance.
>    - More complex agent designs—such as domain-specific prompting, memory modules, or test-time scaling—can indeed improve results in VitaBench. Our Pass@4 experiments already show that perhaps simple majority voting may yield gains. While these techniques are valuable in practice, we avoid integrating them into the benchmark **to maintain clear, model-level comparability**. We appreciate your understanding!
>
> ---
>
> **Reference**
>
> [1] Tau2-Bench: Evaluating Conversational Agents in a Dual-Control Environment, arXiv: 2506.07982
>
> [2] https://openai.com/index/api-prompt-caching
>
> [3] SWE-bench: Can Language Models Resolve Real-World GitHub Issues? ICLR 2024
>
> [4] OpenHands: An Open Platform for AI Software Developers as Generalist Agents, ICLR 2025

---

> > ### Comment · Reviewer_jGAh · 2025-11-25
> > **Reply**
> >
> > Thank you for the authors' replies, which have addressed all my concerns.

---

> ### Author Response · Authors · 2025-11-26
> **Thank You for Your Follow-Up and Consideration**
>
> Dear Reviewer jGAh,
>
> Thank you very much for your follow-up and for confirming that our responses have addressed your concerns. We truly appreciate your time and constructive feedback throughout the process.
>
> If the clarifications and updates meet your expectations, we would be grateful if you could consider reflecting this in your final rating or confidence. Thank you again for your consideration and for helping us improve the work.
>
> Sincerely,
>
> Authors of VitaBench

---

### Author Response · Authors · 2025-11-24
**Overall Response to All Reviewers**

Dear Reviewers,

We are very grateful for your recognition of our work (e.g., the **systematic formalization** of agentic task complexity, the **realism** of benchmark design, and the **comprehensiveness** of experiments), which is a great encouragement to us.

We also appreciate your professional and constructive suggestions, and have revised our manuscript to address all reviewers' concerns. **All modifications in the revised manuscript are highlighted in blue**. In this overall response, we provide a summary of the key actions we have undertaken.

---

## Clarifications

1. **Positioning VitaBench as an evaluation of foundation models rather than agentic systems.** (Reviewer jGAh & zsYD)
   - Our goal is to assess the intrinsic agentic and tool-use capabilities of foundation LLMs, rather than the performance of specific agent frameworks. To avoid confounding effects from heterogeneous agent stacks, we adopt a minimal and widely used function-calling setup that enables clean and directly comparable model-level evaluation.

2. **Design considerations for realism and stability in the user simulator.** (Reviewer Bj6N)
   - While human involvement is impractical at scale, our simulator incorporates structured user personas, explicit behavioral rules, and calibrated intent expressions. These components jointly enhance behavioral realism while keeping the interaction dynamics stable and reproducible.

3. **Cultural fairness in VitaBench and avoidance of culture-dependent knowledge.** (Reviewer Bj6N)
   - Although evaluations operate within a limited cultural context, large-scale multilingual pretraining has substantially reduced regional disparities among modern LLMs. In addition, VitaBench avoids tasks requiring culture-dependent knowledge and supplements necessary background information when relevant.

4. **Operation of the sliding-window evaluator and its dynamic state tracking.** (Reviewer boYB)
   - The evaluator applies binary, fine-grained rubrics to overlapping windows and maintains a dynamically updated state vector across the trajectory. When a previously satisfied criterion becomes invalidated, the corresponding state is reset.

5. **Potential use of VitaBench for training and future model improvement.** (Reviewer zsYD)
   - Although designed primarily for evaluation, VitaBench offers a complete and extensible framework—including a runnable environment, an interactive user simulator, a reliable evaluator, and a systematic formalization of agentic task complexity—that can support data generation or RL training if needed.

## Supplementary Experiments

1. **Analyzing potential bias in different user simulators and the evaluator.** (Reviewer Bj6N)
   - We added an analysis in **Section 5.1** comparing model performance under different user simulators. The results show no explicit or implicit cooperative tendencies with models of the same family. We also clarified that, due to our objective and binary rubric design, the evaluator does not introduce systematic bias.

2. **Evaluating the influence of user personas on performance outcomes.** (Reviewer boYB)
   - We added an additional study in **Section 5.2** examining the effects of varying user personas. Different personas introduce meaningful behavioral variability, which is an intended feature of VitaBench.


## Others

1. We added the full list of tools in **Appendix C** to improve transparency of the environment setup.
2. We added cost and latency statistics of two representative models in **Appendix D**.
3. We added a brief clarification of “score” in the **Table 5** caption.
4. We added several wording refinements throughout the paper to improve clarity, such as replacing *LLM-based Agents* with *Agentic Models*.

---

Thank you again for taking the time to read our response. We hope this addresses your concerns and welcome any further questions or feedback. If our revisions meet your expectations, **we kindly ask for consideration in adjusting your scores accordingly.**

Best regards,

Authors of VitaBench

---

### Author Response · Authors · 2025-12-02
**For AC, SAC and PC: Summary of Rebuttal Process**

Dear Area Chairs, Senior Area Chairs, and Program Chairs,

We hope this message finds you well. To help reduce your reviewing workload, we are writing to provide a summary of the rebuttal process for our paper (Submission 5470: VitaBench).

---

1. **Reviewer jGAh:** Rating: 6 / Confidence: 3
   - **Primary concerns in the review (12 Nov):** cost and latency analysis, and why we only evaluate foundation models using a minimal function-calling agent.
   - **Our response (17 Nov):** we added concrete cost statistics and clarified that VitaBench aims to probe intrinsic model-level agentic abilities under a minimal, widely used design.
   - **Reviewer follow-up (26 Nov)**: confirmed all concerns were **resolved** and maintained a positive rating.
2. **Reviewer Bj6N:** Rating: 4 / Confidence: 4
   - **Primary concerns in the review (12 Nov)**: potential LLM–LLM evaluation biases, realism of the user simulator, and cultural specificity.
   - **Our response (17 Nov):** we provided cross-model bias checks, explained the stability–realism tradeoff in simulator design, and showed that cultural dependence is minimized both by design and by findings from existing works.
   - **No follow-up** from the reviewer until 28 Nov.
3. **Reviewer boYB:** Rating: 6 / Confidence: 3 → 4
   - **Primary concerns in the review (12 Nov)**: novelty beyond existing benchmarks, evaluator design, and persona effects.
   - **Our response (17 Nov):** we highlighted the benchmark’s novelty, including the systematic formalization of Agentic Task Complexity and the design of VitaBench itself, clarified the implementation details of the evaluator, and added experiments quantifying persona influence.
   - **Reviewer follow-up (26 Nov)**: confirmed that all concerns were **resolved**, maintained a positive rating, and **increased** confidence.
4. **Reviewer zsYD:** Rating: 6 / Confidence: 3
   - **Primary concerns in the review (12 Nov):** evaluation limited to foundation models and support for adaptive agent systems.
   - **Our response (17 Nov):** we clarified that the benchmark targets intrinsic model abilities under minimal design, and highlighted that VitaBench still supports data generation and RL training for developing adaptive agents.
   - **Reviewer follow-up (27 Nov)**: confirmed that all concerns were **resolved** and maintained a positive rating.
5. **Overall Response to All Reviewers (25 Nov)**: we revised the paper and summarized the main updates
   - **Clarifications:** Addressed several key questions.
   - **Supplementary Experiments:** Added additional experiment requested by reviewers.
   - **Others:** Added details and information to enhance readability.

---

We also deeply value the reviewers’ thoughtful evaluations, which highlight several strengths of our works:

1. **Clear three-dimension framework** that systematically formalizes agent task complexity and supports cross-scenario composition (Reviewer jGAh, zsYD)
2. **Realistic, real-world–grounded benchmark** with authentic platform data and tasks, and strong motivation for evaluating long-horizon, challenging agent tasks (Reviewer jGAh, Bj6N, boYB, zsYD)
3. **Practical and reliable benchmark components**—including the rubric-based sliding-window evaluator, user simulator, and others—with demonstrated human agreement and robustness (Reviewer jGAh, Bj6N, boYB)
4. **Clear writing and comprehensive experiments** revealing benchmark difficulty with detailed analyses across three complexity dimensions (Reviewer jGAh, boYB, zsYD)

---

In summary, the reviewers provided valuable insights and suggestions, and we responded accordingly. During the Author–Reviewer Discussion phase, **3 out of 4 reviewers provided positive feedback** and confirmed that we addressed their concerns. For Reviewer Bj6N, who has not yet responded, we have likewise provided detailed clarifications and conducted supplementary experiments to address all raised points.

We have revised our manuscript based on the rebuttal content to further strengthen our work (**highlighted in blue**). We hope that VitaBench can provide insights and contribute to the broader development of AI agents.

We sincerely hope that this summary assists you in making a fair and well-informed decision. Thank you for your time and consideration!

Best regards,
All Authors of Submission 5470

---

### Meta-Review · Area_Chair_UKw6 · 2026-01-11

**Summary:**

Reviewer jGAh (score 6)
had questions about the cost and latency across models, they also wanted to see how different agents would perform on this benchmark.

Reviewer Bj6N (score 4)
had concerns about using LLMs for evaluation of the task completion. They also commented on the fact that agents in this benchmark are expected to follow complex but highly specific, unambiguous specifications, as opposed to humans who typically engage in or expect ambiguous decision making. So the benchmark is more suited for evaluating agents rather than realistic agentic tasks.

Reviewer boYB (score 6)
commented that there exist many similar benchmarks and evaluation methodology that evaluates the task over a sliding window (as opposed to a holistic evaluation of the entire trajectory).

Reviewer zsYD (score 6)
wanted to see more LLM models being evaluated on the benchmark. They also thought that the benchmark only evaluates static agents that do not learn from the experiences over time.

This paper develops a new benchmark that simulates a user, an agent that performs the task and an evaluator that checks the result. The paper shows results on real-world-like tasks such as ordering food or booking a hotel room. The reviewers mostly had questions about the evaluation methodology. The 4 reviewers are borderline positive about the paper. This paper presents a great beginning. While updating the manuscript, the authors are encouraged to incorporate the comments by the reviewers. For example, it is necessary to focus on and argue for a precise scientific question that is addressed by this benchmark. If the question is “what constitutes task complexity” (as the paper argues in the introduction) then the present manuscript does not shed any specific light upon it. Some important comments raised in the reviews, e.g., what exactly this benchmark evaluates (since all of them are simulated), using an appropriate evaluation methodology (for such graph/specifications based tasks a sliding window is a poor substitute), and that this benchmark does not evaluate how agents would learn from these interactions (this is a wonderful question to focus on for future work).

**Reviewer Concerns:**

Reviewer jGAh
The rebuttal provided results on cost analysis that will be included in the paper. The authors also argued that the paper seeks to evaluate basic function-calling agents across broad domains instead of more elaborate design patterns such as Open Hands that may be more effective in specific domains like coding.

Reviewer Bj6N
The authors provided additional experiments where the user simulator is changed to Claude Sonnet, the results are similar to the ones in the paper with GPT-4.1. The second comment about modeling/addressing emergent human behavioral specifications is perhaps beyond the scope of the present paper.

Reviewer boYB
The authors argued that the new benchmark is indeed novel because it focuses on reasoning, tool use and interaction complexity (something that none of the existing benchmarks do). They made a similar point about evaluation, namely that many benchmarks such as Tau Bench only evaluate the final state and not even the trajectory.

Reviewer zsYD
The response was similar to the one to the first reviewer.

**Reviewer Scores:**

Reviewer jGAh
The score was updated after the rebuttal, it is currently 6 (accept).

Reviewer Bj6N
The score would likely be updated to a 6 (accept).

Reviewer boYB
The score was updated to a 6 (accept).

Reviewer zsYD
The score was kept the same, to a 6 (accept).

---

### Decision · Program_Chairs · 2026-01-26

Accept (Poster)